# Robust, coherent, and synchronized circadian clock-controlled oscillations along *Anabaena* filaments

Rinat Arbel-Goren[1], Valentina Buonfiglio[2], Francesca Di Patti[3], Sergio Camargo[1], Anna Zhitnitsky[1], Ana Valladares[4], Enrique Flores[4], Antonia Herrero[4], Duccio Fanelli[2], Joel Stavans[1]*

[1]Department of Physics of Complex Systems, Weizmann Institute of Science, Rehovot, Israel; [2]Dipartimento di Fisica e Astronomia, Università di Firenze, INFN and CSDC, Sesto Fiorentino, Italy; [3]Consiglio Nazionale delle Ricerche, Istituto dei Sistemi Complessi, Sesto Fiorentino, Italy; [4]Instituto de Bioquímica Vegetal y Fotosíntesis, CSIC and Universidad de Sevilla, Sevilla, Spain

**Abstract** Circadian clocks display remarkable reliability despite significant stochasticity in biomolecular reactions. We study the dynamics of a circadian clock-controlled gene at the individual cell level in *Anabaena* sp. PCC 7120, a multicellular filamentous cyanobacterium. We found significant synchronization and spatial coherence along filaments, clock coupling due to cell-cell communication, and gating of the cell cycle. Furthermore, we observed low-amplitude circadian oscillatory transcription of *kai* genes encoding the post-transcriptional core oscillatory circuit and high-amplitude oscillations of *rpaA* coding for the master regulator transducing the core clock output. Transcriptional oscillations of *rpaA* suggest an additional level of regulation. A stochastic one-dimensional toy model of coupled clock cores and their phosphorylation states shows that demographic noise can seed stochastic oscillations outside the region where deterministic limit cycles with circadian periods occur. The model reproduces the observed spatio-temporal coherence along filaments and provides a robust description of coupled circadian clocks in a multicellular organism.

*For correspondence:
joel.stavans@weizmann.ac.il

**Competing interests:** The authors declare that no competing interests exist.

## Introduction

Endogenous circadian clocks allow the alignment of cellular physiology with diurnal light/darkness cycles on Earth, endowing organisms, from unicellular cyanobacteria to multicellular plants and mammals, with a selective fitness advantage (*Cohen and Golden, 2015*). Significant progress has been achieved in understanding circadian clock architectures and function in cyanobacteria, which are arguably the simplest organisms exhibiting self-sustained oscillations (*Hasty et al., 2010*; *Rust et al., 2007*; *Lambert et al., 2016*; *Dong et al., 2010*; *Teng et al., 2013*; *Gan and O'Shea, 2017*). The molecular mechanisms behind autonomous circadian clocks have been elucidated primarily in the unicellular *Synechococcus elongatus*. These investigations have shown that the core of the circadian clock consists of three proteins, KaiA, KaiB, and KaiC (*Ishiura et al., 1998*), whose oscillating behavior can be reconstituted *in vitro* (*Nakajima et al., 2005*). The basic mechanism behind the clock is based on the stimulation of KaiC autophosphorylation by the binding of KaiA and the autodephosphorylation that ensues when KaiB binds to KaiC, blocking KaiA's stimulatory function. A salient feature of the circadian clock in *Synechococcus* is the high temporal precision it can exhibit, despite the fact that biochemical reactions in a cell are stochastic events and that clock components may be subject to variations in molecular copy numbers between cells, variations known as demographic noise (*Tsimring, 2014*). Many studies have addressed the robustness of circadian rhythms to

demographic noise in Kai proteins (*Mihalcescu et al., 2004*; *Chabot et al., 2007*; *Teng et al., 2013*; *Pittayakanchit et al., 2018*; *Chew et al., 2018*), but copy number variations in KaiC phosphoforms, which impact directly on clock function, have not been previously considered.

The multicellular character of higher organisms and of some cyanobacterial species (*Herrero et al., 2016*) naturally prompts the question of how do ensembles of noisy circadian clocks perform in a multicellular organismal setting. Theoretical considerations have motivated the notion that reliable collective oscillations may result from the coupling of 'sloppy', noisy clocks (*Enright, 1980*). It has been suggested that coupling of circadian clocks in unicellular organisms by quorum sensing interactions may result in emergent synchronization (*Garcia-Ojalvo et al., 2004*), and experimental evidence in support of this notion has been reported in a synthetic system (*Danino et al., 2010*).

*Anabaena* sp. strain PCC 7120 (henceforth *Anabaena*) is a multicellular cyanobacterium in which cells are arranged in a one-dimensional configuration, with local, nearest-neighbor cell-cell coupling through septal junctions (*Herrero et al., 2016*). Evidence of coupling of metabolic pathways along a filament due to cell-cell communication has been reported (*Mullineaux et al., 2008*). In contrast to *Synechococcus*, information about the mechanism of the circadian clock of *Anabaena* is scant. Sequence BLAST analysis has shown that *Anabaena* contains homologs of the *kaiA*, *kaiB*, and *kaiC* genes of *Synechococcus*, and structural studies indicate that the interactions between the respective proteins are similar (*Garces et al., 2004*). Furthermore, *Anabaena* also contains factors coupling the Kai post-transcriptional oscillator to input signals and to output factors such as RpaA, CikA, and SasA that couple the clock to the genes it regulates (*Schmelling et al., 2017*). The roles of these genes in *Anabaena* remain to be elucidated. A first characterization of the circadian rhythms in bulk cultures of *Anabaena* has shown that the clock is autonomous, running freely under constant light conditions following exposure to two 12 hr light-dark cycles, similarly to *Synechococcus* (*Kushige et al., 2013*). However, this study also showed that, in contrast to *Synechococcus*, none of the *kai* genes are expressed with a large amplitude. Nonetheless, about 80 *kai*-controlled genes that exhibit high-amplitude circadian oscillations were identified using DNA microarray analysis, a behavior that was abolished in a *kaiABC* deletion mutant.

Here, we present an experimental and theoretical study of circadian clocks in multicellular *Anabaena*. Its one-dimensional character allowed us to follow clocks in each and every cell along a filament, and shed light on the interplay between demographic noise and cell-cell communication in setting synchrony and spatial coherence along filaments. In our experiments, we followed *in vivo* the output of clocks in individual cells by monitoring the expression from the promoter of *pecB*, a clock-controlled gene of high-amplitude oscillations (*Kushige et al., 2013*). This gene is part of the *pecBA-CEF* operon and codes for the beta subunits of phycoerythrocyanin, a structural component of the phycobilisome rod that plays a major role in light harvesting for photosynthesis (*Swanson et al., 1992*). On the theoretical side, we first incorporated the effects of demographic noise into a three-component model of a single clock describing the phosphorylation states of KaiC, as in *Synechococcus*. Next, we extended this single-cell stochastic model to describe a one-dimensional array of coupled clocks, as in multicellular *Anabaena*, allowing us also to analyze the spatio-temporal coherence properties of noisy oscillations in *Anabaena* filaments.

## Results

### Circadian clocks of individual cells in growing filaments are highly synchronized

We followed circadian oscillations in *Anabaena* from a chromosomal *gfp* fusion to the N-terminus of the clock-controlled protein, PecB, here denoted as $P_{pecB-gfp}$ at the level of individual cells along filaments. Prior to and during the experiments, filaments were grown under constant light conditions. Typical images of wild-type (WT) filaments at different time points are shown in *Figure 1A* (see also *Figure 1*, *Video 1*). One salient feature in the images is significant synchrony, that is, cellular oscillations progressed in individual cells along filaments together and with a similar period. The images in *Figure 1A* correspond to successive maxima and minima of the mean cell fluorescence intensity, μ, as a function of time, which we plot in *Figure 2A*. Similar experiments with a strain in which the *kaiABC* genes were deleted (Δ*kaiABC*) resulted in a low-level, non-oscillatory signal (*Figure 2A*),

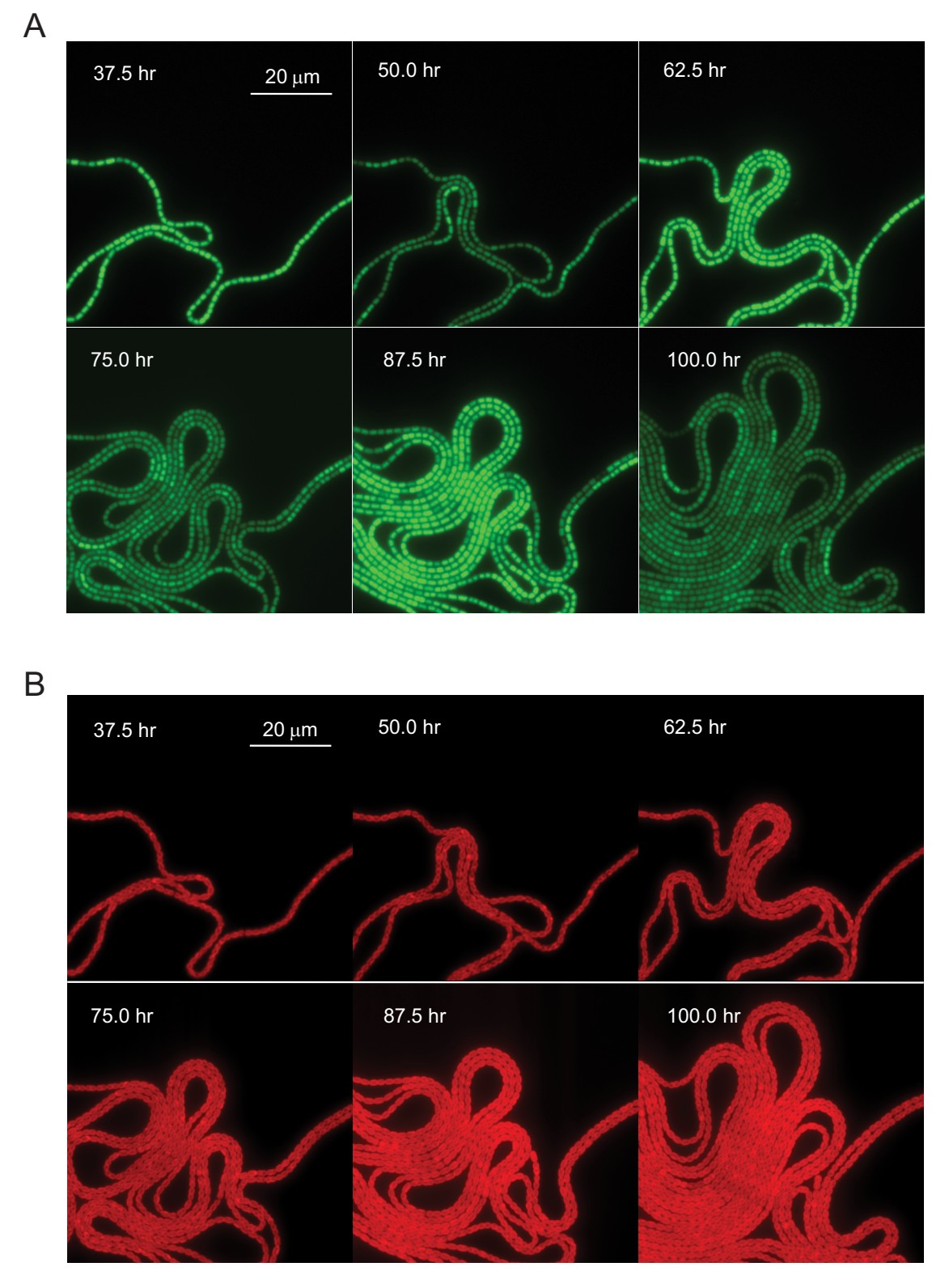

**Figure 1.** Circadian oscillation in *Anabaena*. (A) GFP fluorescence in a filament of an *Anabaena* strain bearing a $P_{pecB-gfp}$ promoter fusion, growing under nitrogen-replete conditions. The snapshots were chosen near maxima and minima of the circadian oscillations. (B) Autofluorescence as a function of time in *Anabaena*. Snapshots correspond to those in (A), and time 0 corresponds to the time at which filaments were placed in a device for microscope observation (for details, see Materials and methods). For a time-lapse movie, see *Video 1* (taken over 6 days).

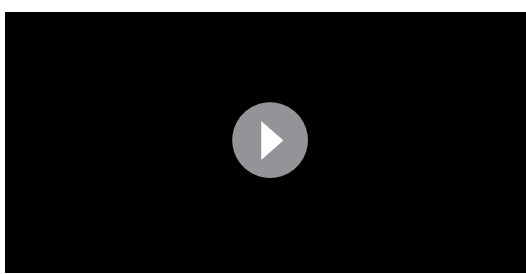

**Video 1.** Real-time expression of a clock-controlled gene and filament autofluorescence during circadian oscillations in WT *Anabaena*.
https://elifesciences.org/articles/64348#video1

confirming the regulation of *pecB* expression by the circadian clock genes. As a control, we monitored expression from the promoter of *hetR*, a gene coding for the master regulator of development in *Anabaena*. The expression from the *hetR* promoter $P_{hetR-gfp}$ did not exhibit an oscillatory behavior (*Figure 2A*), a result consistent with previous microarray experiments (*Kushige et al., 2013*). This does not preclude a possible interaction between the circadian clock and differentiation. On the other hand, the autofluorescence from photosynthetic pigments did not display oscillatory behavior (see *Figure 1B*, *Figure 2A*).

To characterize quantitatively the degree of synchronization between cellular clocks along a filament, we used the synchronization index $R$ (*Garcia-Ojalvo et al., 2004*) (Materials and methods), which can be readily calculated from the measured fluorescence intensity in each cell and which varies between 0 (no synchronization) and 1 (full synchronization). To compute $R$, several cells along a filament were selected, and their fluorescence intensity was followed over a full period of oscillation. Contiguous cells, which include sister cells from the same mother, are highly synchronized, as shown by the large value of $R$ $(0.89 \pm 0.04)$ (*Table 1*). The value of $R$ for cells initially separated by intervals of 10 other cells – chosen in an attempt to avoid initial correlations between their clocks – was significantly indistinguishable from that computed for contiguous cells, underscoring the large degree of synchronization of clocks along a filament.

To test the extent to which different filaments are synchronized under the same conditions, we measured the average fluorescence intensity per cell in different filaments as a function of time (*Figure 2B*). The expression from $P_{pecB-gfp}$ in different filaments oscillated nearly in phase mainly during the first oscillations. To evaluate quantitatively the degree of phase synchronization between filaments, we calculated $R$ by choosing one cell per filament in a number of filaments measured simultaneously (Materials and methods). We obtained $R = 0.75 \pm 0.04$, a value that is significantly smaller than that obtained for cells within the same filament (*Figure 2C*, *Table 1*). We surmise that initial synchronization may be due to phase resetting following the change in conditions, for example, illumination, upon transfer of cells from bulk culture to the microscope for real-time measurements. This change also could account for the decay in fluorescence intensity observed during the first circa 24 hr of our experiments (*Figure 2A, B*). Furthermore, this decay is $P_{pecB-gfp}$-specific, but clock-independent, as it was also observed in filaments of the $\Delta kaiABC$ background (*Figure 2A*).

## Circadian clocks along filaments are coupled by cell-cell communication

Another salient feature in the snapshots of *Figure 1A* is the high spatial coherence of the expression from $P_{pecB-gfp}$ along filaments, that is, all cells have nearly the same phase along their circadian cycle. To quantify the extent to which clocks are actually correlated, we calculated the spatial autocorrelation function of fluorescence intensity $g$ as a function of distance along a filament (see Materials and methods). We found that $g$ decays to zero for separations of five cells or more (*Figure 2D*).

To evaluate the contribution of phase inheritance following cell division to the observed autocorrelation, a simulated filament was generated from each measured filament by dividing each cell into two for three generations, partitioning the fluorescence of a mother cell binomially between the two daughters, and then multiplying the results by two, in order to conserve the average fluorescence per cell. The autocorrelation functions of these simulated filaments were then computed and averaged. The resulting mean autocorrelation (*Figure 2D*, magenta line) decreased to zero already at the second neighbor, suggesting that another factor, for example, cell-cell communication, contributes significantly to the coupling of fluctuations of *pecB* expression along a filament. To support this notion further, we calculated the spatial autocorrelation function of $P_{pecB-gfp}$ expression in filaments

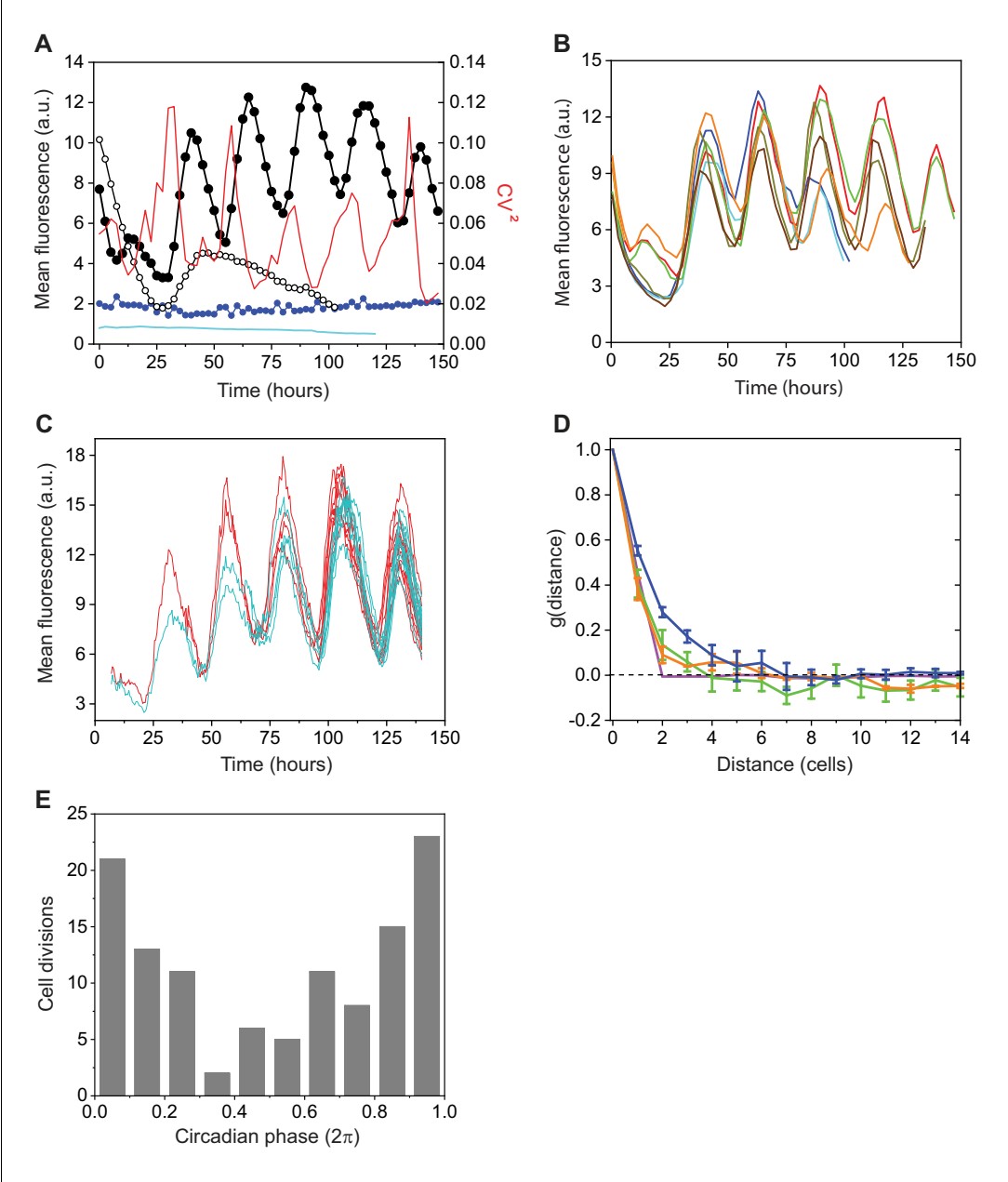

**Figure 2.** Characterization of a clock-controlled gene in *Anabaena*. (A) Average cell fluorescence intensity from $P_{pecB-gfp}$ in a filament as a function of time for a wild-type genetic background (full black circles) and for a $\Delta kaiABC$ background (empty black circles); intensity of autofluorescence as a function of time (blue circles); average cell fluorescence intensity from $P_{hetR-gfp}$ (cyan line); and temporal dependence of the cell-cell variability $CV^2$ in expression of $P_{pecB-gfp}$ (red line). Data were taken from at least 50 contiguous cells along a filament. (B) Average fluorescence intensity as a function of time of filaments in different fields of view from the same experimental run. Each trace was obtained from at least 50 contiguous cells along each filament. (C) Expression from a $P_{pecB-gfp}$ fusion in the lineages of two contiguous cells as a function of time. (D) Average spatial autocorrelation function of $P_{pecB-gfp}$ expression along filaments of wild-type (blue), $\Delta sepJ/\Delta fraCD$ (green), and $\Delta kaiABC$ (orange) genetic backgrounds. Error bars represent standard errors. Magenta line: contribution to the spatial autocorrelation function of fluctuations from the wild-type data set, induced by binomial partitioning of molecules between daughter cells, following each of three consecutive cell divisions. Prior to divisions, the cell order in each filament was reshuffled. (E) Histogram of the phase of cell-division events along the circadian cycles, with 0 and $2\pi$ denoting two consecutive minima, from two independent experiments. For additional data similar to (A) and (C), corresponding to filaments of the *sepJ/fraCD* genetic background, see *Figure 2—figure supplement 1*.

The online version of this article includes the following figure supplement(s) for figure 2:

**Figure supplement 1.** Effects of perturbation of cell-cell communication on the expression of PpecB−gfp.

**Table 1.** Synchronization of expression of a clock-controlled gene in cells within and between *Anabaena* filaments.

The synchronization index $R$ for strains with the indicated genotypes (Materials and methods) was measured from the fluorescence intensities of $P_{pecB-gfp}$ expression in the same cells followed over a full circadian period in a filament, either in clusters of contiguous cells (contiguous) or for cells separated by intervals of 10 cells (separate). To measure synchronization between filaments, $R$ was computed from about 10 cells, each belonging to a different filament. For each genetic background, the mean and standard error of the mean (SEM) of $R$ was determined from a number of independent repeats (***Rust et al., 2007***; ***Lambert et al., 2016***; ***Dong et al., 2010***; ***Teng et al., 2013***), carried out in $n$ different experimental runs. Significance (p-value) in interstrain comparisons was established by the Mann–Whitney U-test, and * represents rejection of the null hypothesis that samples come from distributions with equal medians. WT: wild type.

| Genotype | Cell cluster | R (mean ± SEM) | n | Comparison with strain | p-Value |
|---|---|---|---|---|---|
| WT | Contiguous | 0.89 ± 0.04 | 3 | WT (separate) | 0.117 |
| WT | Separate | 0.85 ± 0.01 | 2 | | |
| WT | Different filaments | 0.75 ± 0.04 | 2 | WT (separate) | 0.026* |
| $\Delta sepJ\Delta fraCD$ | Contiguous | 0.73 ± 0.05 | 4 | WT (contiguous) | 0.001* |
| $\Delta kaiABC$ | Contiguous | 0.71 ± 0.03 | 3 | WT | 0.001* |

of a $\Delta sepJ/\Delta fraCD$ strain in which genes coding for three septal proteins SepJ, FraC, and FraD involved in cell-cell communication were deleted (***Nürnberg et al., 2015***). Circadian oscillations in expression from $P_{pecB-gfp}$ in filaments of this strain were observed (***Figure 2—figure supplement 1A***). The resulting spatial autocorrelation function decreased over significantly shorter lengthscales (about two cells) than the WT (***Figure 2D***). Therefore, we calculated the synchronization index between contiguous cells in this genetic background and found that $R$ was significantly smaller than that measured for contiguous cells within filaments of the WT background, but comparable to that obtained for cells in different filaments (***Table 1***). Consistently with this smaller value of R, traces of individual cells and their respective lineages were considerably more noisy (see ***Figure 2—figure supplement 1B***) than lineages in WT filaments (***Figure 2C***). These findings indicate that the correlation in $P_{pecB-gfp}$ expression is due primarily to significant coupling between the clocks in neighboring cells due to cell-cell communication. Of note, the value of $R$ calculated for a $\Delta kaiABC$ background, in which $P_{pecB-gfp}$ expression is clock-independent, was similar to that for $\Delta sepJ/\Delta fraCD$, in which cell-cell communication is impaired (***Table 1***).

## Cell-cell variability oscillates out of phase with the circadian rhythm

Variations in gene expression between cells along a filament may limit both synchrony and spatial coherence. These variations are evident in ***Figure 1A*** even though their amplitude is small relative to the clock-modulated activity of $P_{pecB-gfp}$. To quantify these variations, we calculated the square of the coefficient of variation $CV = \sigma/\mu$, where $\sigma$ denotes the standard deviation of the fluorescence intensity of contiguous cells along a filament. A plot of $CV^2$ as a function of time showed that the cell-cell variability itself displays oscillatory behavior, attaining maxima approximately in the middle of periods during which the mean fluorescence intensity increases (***Figure 2A***).

## Coupling between cell cycle and clocks

In a cellular setting, circadian and cell cycle oscillations take place concurrently. In a variety of organisms, from prokaryotes to mammals, the circadian clock has been observed to gate cell division, enabling cell division to take place at some phases of the circadian cycle but inhibiting at others (***Mori, 2009***; ***Yang et al., 2010***). To test whether the cell cycle and circadian clocks are coupled in *Anabaena* cells, we monitored the time at which cell division takes place along the circadian cycle in individual cell traces, under conditions of constant illumination. The fluorescence intensity traces of two contiguous ancestor cells bearing the $P_{pecB-gfp}$ fusion and their respective lineage are shown in

*Figure 2C*. In *Figure 2E*, we show a histogram of the timing of cell division events as a function of the phase at which they occur along the circadian clock, obtained from traces similar to those in *Figure 2C*. Far from being equiprobable along the circadian cycle, cell division events showed a marked tendency to occur near the beginning or the end of a circadian cycle (minima in μ) as for *Synechococcus* (*Yang et al., 2010*). *Figure 2C* also shows that the clock phase was faithfully inherited by any two daughters following cell division.

## Transcriptional oscillations of *kai* genes and the master transducer regulator *rpaA*

Previous northern blot measurements and microarray analysis of *kai* genes showed no reliable, high-amplitude oscillatory expression of any of the *kai* genes in *Anabaena*. To study with higher sensitivity the expression of *kai* genes, we carried out real-time quantitative polymerase chain reaction (RT-qPCR) measurements of WT and Δ*kaiABC* strains in bulk cultures. Since the value of the $R$ index was smaller between filaments than within (*Table 1*), the experiments were carried out under constant light conditions following two 12 hr/12 hr light-dark cycles, to enhance synchronization (see Materials and methods). The relative expression of the three *kai* genes indeed showed noisy, low-amplitude temporal modulations (*Figure 3A*). The oscillations were largely in phase, and transcription occurred mainly during subjective day. To assess quantitatively the extent of coordination, we calculated the synchronization index $R$ for the different pairs and obtained $R_{A,B} = 0.85 \pm 0.05$, $R_{A,C} = 0.79 \pm 0.09$, and $R_{B,C} = 0.87 \pm 0.06$. Error bars were obtained by bootstrap methods (Materials and methods). Since the differences between these values are not significant, we conclude that transcription of the three genes is coordinated. In fact, the value of $R$ evaluated for the three genes was $0.78 \pm 0.08$.

To expose an underlying oscillatory behavior in the *kai* genes data and support the notion that the oscillations are circadian, we applied persistent homology methods (*Pereira and de Mello, 2015*; *Otter et al., 2017*) to two-dimensional phase portraits of their respective time series (see Appendix 1 – supplemental methods). The delay $\tau$ for each phase portrait corresponds to the first minimum of the auto-mutual information of the time series (*Fraser and Swinney, 1986*), and for a periodic, nearly sinusoidal signal, corresponds to a quarter of the signal's period (*Kennedy et al., 2018*). We obtained $\tau = 7.1 \pm 1.2$ hr, $\tau = 6.7 \pm 1.2$ hr, and $\tau = 7.3 \pm 1.1$ ($n \geq 3$) hr for *kaiA*, *kaiB*, and *kaiC*, respectively, all consistent with a circadian period of oscillation (Appendix 1). A Vietoris–Rips filtration of the clouds of points in the phase portraits (Appendix 1) indeed showed evidence for a persistent cycle in the transcription of each of the *kai* genes (*Figure 3—figure supplement 1*).

To shed light on how the state of the clock is relayed to the genes it controls, for example, *pecB*, we measured the relative expression of *rpaA* in WT and Δ*kaiABC* strains by RT-qPCR. RpaA has been shown to transduce the phosphorylation state of KaiC to clock-controlled genes in other cyanobacteria (*Markson et al., 2013*; *Iijima et al., 2015*) and is highly conserved (*Schmelling et al., 2017*). The role RpaA plays in the circadian oscillations of *Anabaena* is unknown. We found that the relative expression of *rpaA* displays high-amplitude oscillatory behavior (*Figure 3B*). Furthermore, *pecB* displays oscillatory behavior, with similar amplitude and phase. The oscillatory behavior of both genes was abolished in the Δ*kaiABC* mutant (*Figure 3B, C*). Note that the expression of both *rpaA* and *pecB* peaks during subjective night.

## Incorporating demographic noise into a model of a single clock

In order to develop a model of an array of coupled noisy clocks as in *Anabaena*, we first characterized the spectral properties of uncoupled clocks in individual cells of *Synechococcus*. We carried out experiments following the expression of YFP from the *kaiBC* promoter in single cells of *Synechococcus* growing within patterned gels (*Figure 4A*). Circadian oscillations in the lineages of two sister cells are shown in *Figure 4B* (see also *Figure 5B* for the associated power spectrum). Expression from the *kaiBC* promoter exhibited circadian oscillations with a period of about 25 hr, similar to those observed previously (*Teng et al., 2013*). Our next goal was to generalize a deterministic model for individual clocks in *Synechococcus* (*Rust et al., 2007*) to include the effects of demographic stochasticity. We followed the interaction network depicted in *Figure 5A* (adapted from *Rust et al., 2007*). Our mathematical model takes into account the dynamics of three forms of KaiC, namely, the single-phosphorylated forms S-KaiC (phosphorylated at serine 431), T-KaiC (phosphorylated at threonine 432), and the double-phosphorylated form D-KaiC, while the unphosphorylated

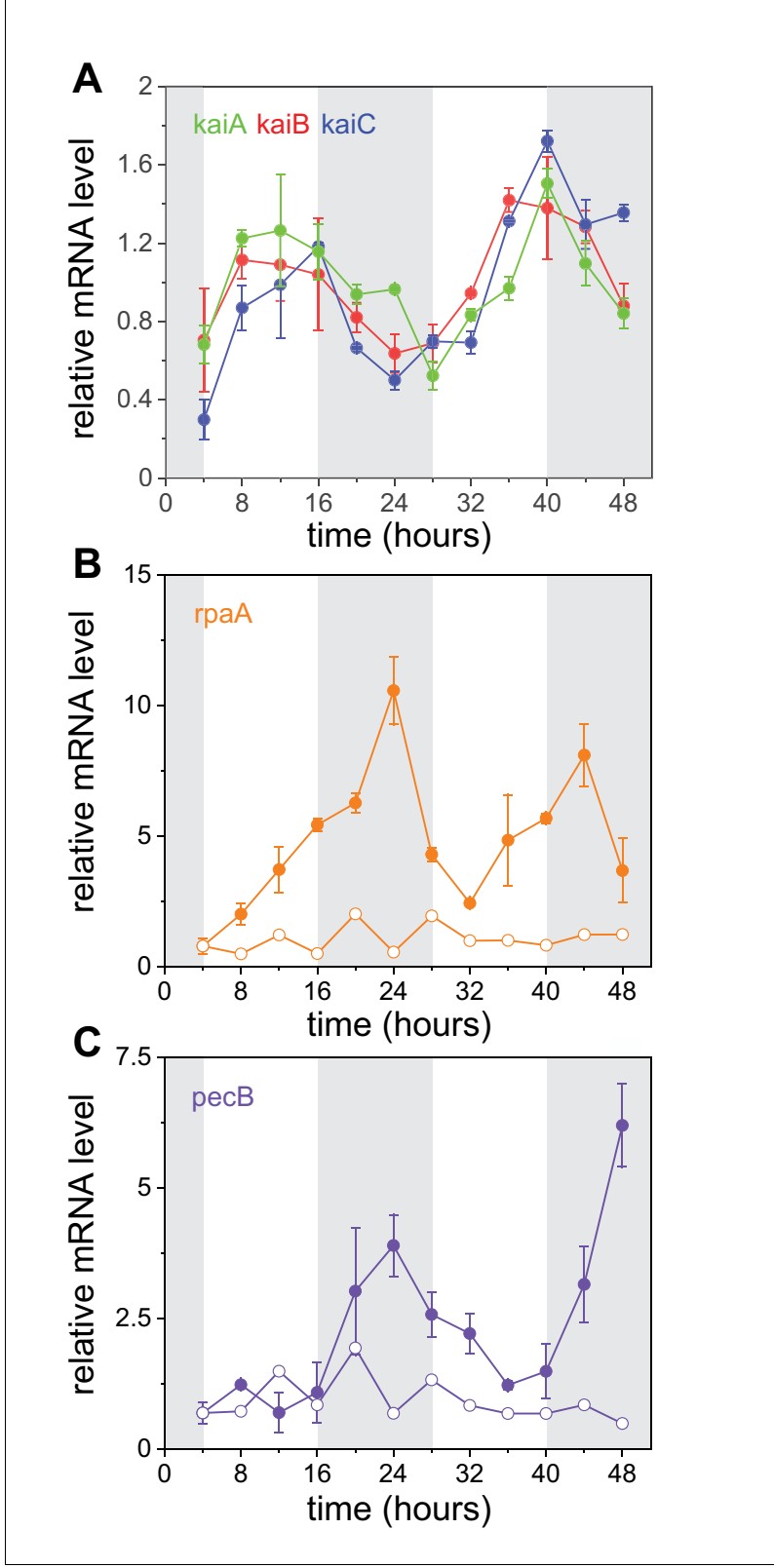

**Figure 3.** Transcriptional oscillations in the core clock genes, *rpaA* and *pecB*. (A) Relative expression of *kaiA* (green), *kaiB* (red), and *kaiC* (blue) as a function of time measured by RT-qPCR (Materials and methods). A persistence homology analysis of these data is presented in *Figure 3—figure supplement 1*. (B, C) Relative expression levels of *rpaA* and *pecB*, respectively, in wild-type (full circles) and Δ*kaiABC* strains (empty circles). *Figure 3 continued on next page*

*Figure 3 continued*

Curves have been normalized by their temporal mean. Error bars represent the standard error of the mean of three independent experiments (see Materials and methods). Gray shades represent periods of subjective night. For additional information about regulatory sequences of the *kaiABC, rpaA, pecB* promoter regions and RpaA binding sites in *Anabaena*, see (C) (*Figure 3—figure supplement 2*).

The online version of this article includes the following figure supplement(s) for figure 3:

**Figure supplement 1.** Persistent homology analysis of periodic behavior in the trascriptionaltime series of kai genes of *Anabaena*.

**Figure supplement 2.** Schematic representation and regulatory sequences of the kaiABC, rpaA,pecB and ftsZ promoter regions in *Anabaena*.

U-KaiC can be deduced from the conservation of the total number of molecules of KaiC. Crucial for the appearance of oscillations is the negative feedback mediated by S-KaiC through inactivation of KaiA via KaiB function. Furthermore, the condition on the active KaiA monomers (see Equation S4 in *Rust et al., 2007*) is modeled here by a continuous function (Appendix 1 – *Equation 6*) that makes analytical progress possible. The parameter $\gamma$ in our nonlinear phosphorylation and dephosphorylation rates $k_{XY}$ for $X, Y = \{U, T, D, S\}$ sets the steepness of the inverted sigmoidal dependence of the rates on KaiA (*Figure 5—figure supplement 1*). The dynamics of the stochastic model is described

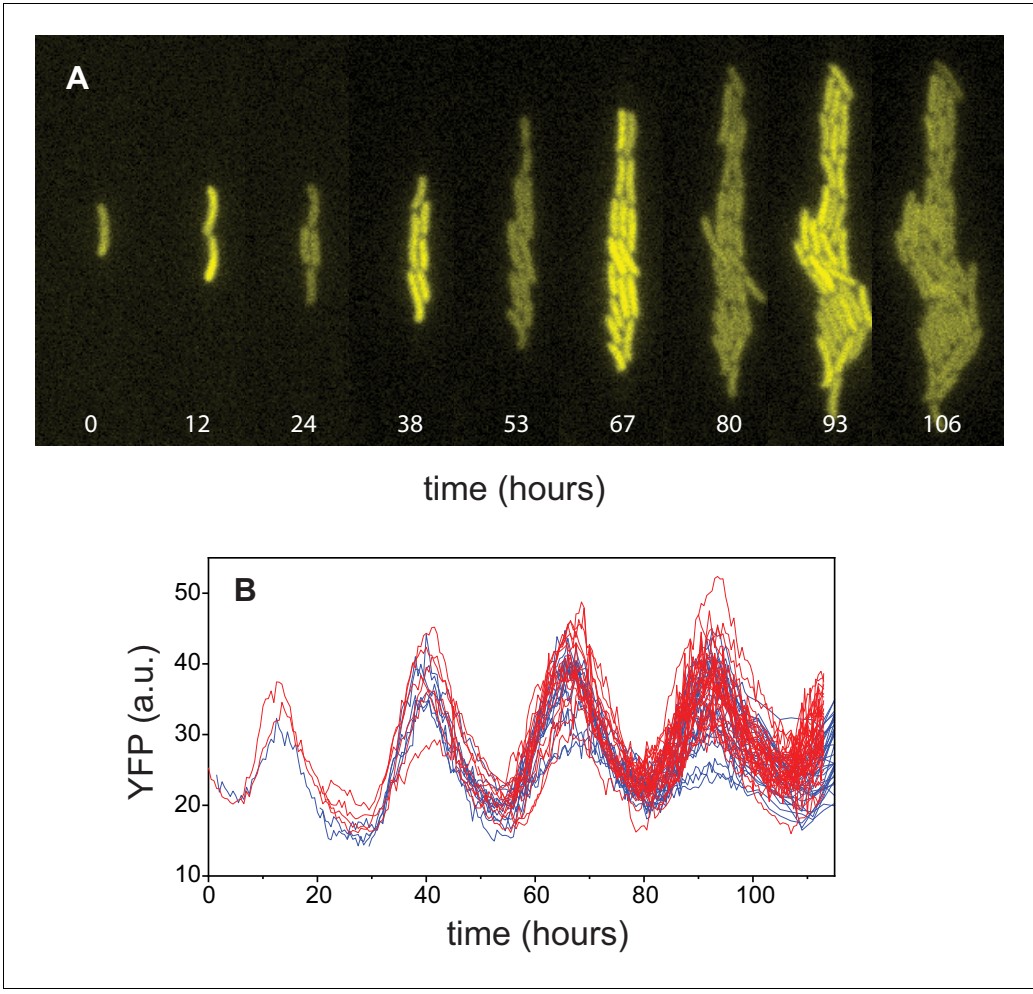

**Figure 4.** Circadian oscillations in *Synechococcus*. (A) Growth and lineage of a cell in patterned agarose, expressing YFP from the *kaiBC* promoter. The snapshots were chosen near maxima and minima of the circadian oscillations. (B) Fluorescence intensity of YFP of individual cells obtained from two independent cell lineages (red and blue).

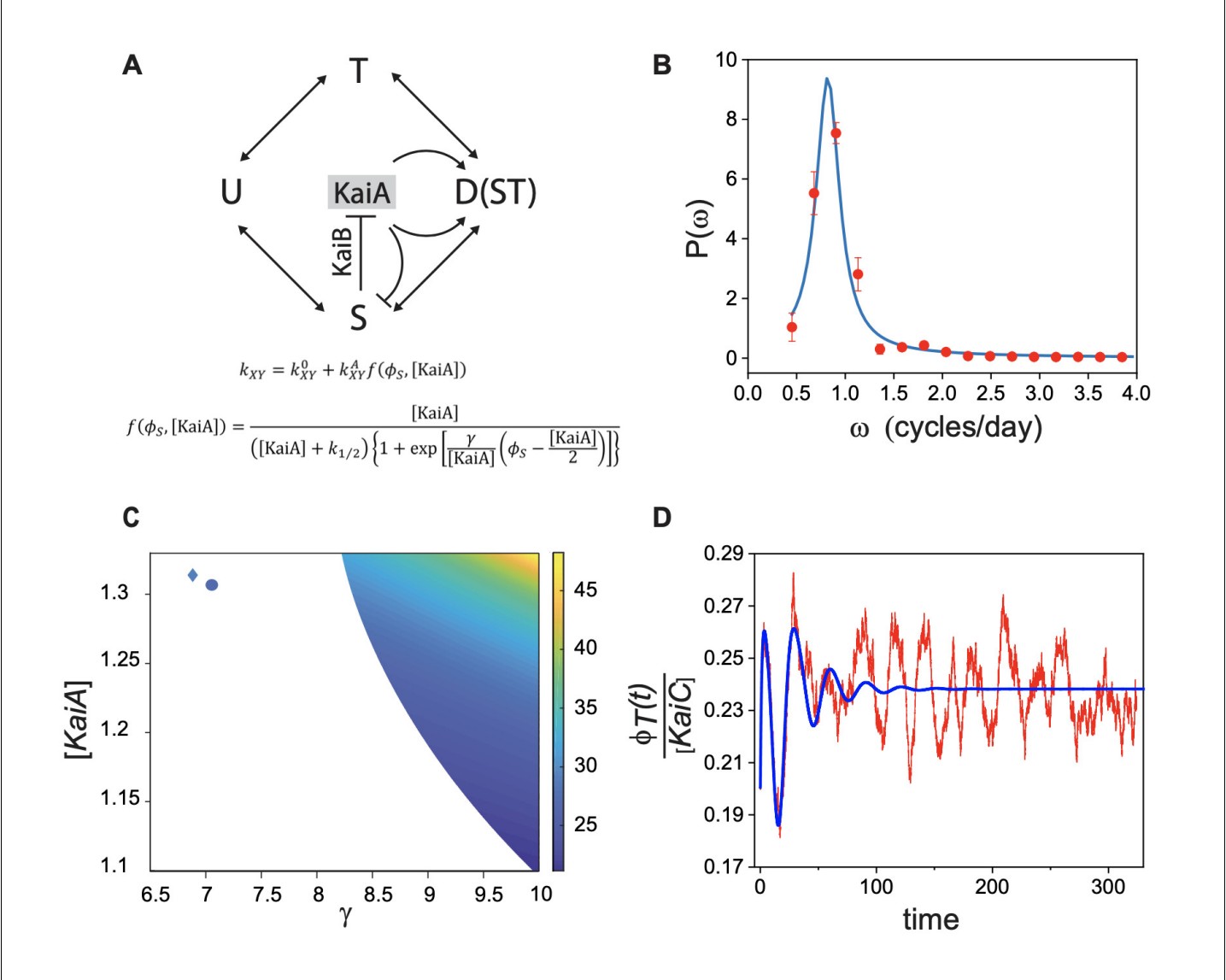

**Figure 5.** Stochastic model for circadian oscillations in *Synechococcus*. (**A**) Schematic representation of interconversion between KaiC phosphoforms modulated by the activity of KaiA in an individual clock. The different phosphoform states of KaiC are denoted by U (unphosphorylated, U-KaiC), T (phosphorylated at threonine, T-KaiC), S (phosphorylated at serine, S-KaiC), and D (phosphorylated at both sites). Arrows denote transitions between the different phosphoforms $X, Y$ with the indicated rates $k_{xy}$. KaiB mediates the inactivation of KaiA by S, as described by the continuous function $f$ (see *Figure 5—figure supplement 1*). (**B**) Average power spectrum of single-cell fluorescence (red symbols) fit to the data with the prediction from the stochastic model (blue line). (**C**) $\gamma$-[KaiA] plane where deterministic limit cycle oscillations in individual clocks occur. The color corresponds to the period of oscillations (in hours). The boundary of the colored region corresponds to a Hopf bifurcation. Note that deterministic oscillations with a circadian period are limited only to a small strip near the stability boundary at the bottom right. The circle identifies the values of $\gamma$ and KaiA that we obtain by fitting experimental power spectra in (**B**). The diamond stands for best fit parameters obtained for *Anabaena* (*Figure 6C*). (**D**) Comparison between damped deterministic oscillations (blue line) and quasi-cycles, both at the circle point outside the region of the deterministic oscillations in (**C**). The online version of this article includes the following figure supplement(s) for figure 5:

**Figure supplement 1.** Typical shape of the nonlinear function f.

by a master equation accounting for discrete variations in molecular copy numbers of phosphoforms instead of deterministic, ordinary differential equations (*Rust et al., 2007*; *Brettschneider et al., 2010*). We then use the van Kampen system-size expansion to carry out a linear noise approximation that yields, to leading order, an extended set of ordinary differential equations for the concentrations of S ($\phi_S = [S - KaiC]$), T ($\phi_T = [T - KaiC]$), and D ($\phi_D = [D - KaiC]$). The analysis of these

equations allows us to determine the region in parameter space within which the model exhibits sustained deterministic oscillations (*Figure 5C*). The parameters of the model have been set to the values determined *in vitro* in *Rust et al., 2007* and reported in *Appendix 1—table 1*, except for $\gamma$ and [KaiA], which were allowed to change freely. Note that deterministic oscillations with a circadian period were limited only to a small strip near the stability boundary. At the next-to-leading order, the expansion allows to evaluate the effects of demographic noise and calculate the power spectrum of fluctuations for each species' abundance due to finite size effects (see Appendix 1). A fit of the theoretical power spectrum to the experimentally measured one provides an adequate interpolation upon adjusting the two fitting parameters, $\gamma$ and [KaiA] (see *Figure 5B*). In drawing the comparison between theory and experiments, we assumed that the fluorescence intensity is an immediate proxy of the phosphoform T-KaiC (*Teng et al., 2013*). The fitted parameters position the system outside the region of deterministic oscillations (circle in *Figure 5C*), suggesting that circadian rhythms can be a manifestation of noise-driven oscillations (*Figure 5D*). Remarkably, the fitted value for $[\mathrm{KaiA}] = 1.308$ μM matches the concentration reported previously ($[\mathrm{KaiA}] = 1.3$ μM; see *Rust et al., 2007*).

## Theoretical model of arrays of coupled noisy clocks

Next, we generalized the single-clock model above to an array of coupled circadian clocks as in *Anabaena*, which is endowed with cell-cell communication via septal proteins (*Herrero et al., 2016*; *Figure 6A*). We postulate that the intercellular transfer of factors such as sugars (*Mullineaux et al., 2008*; *Nürnberg et al., 2015*) may affect the behavior of neighboring clocks, yielding an effective long-ranged coupling between clock inputs across the filament. The interaction is here modeled by an exponential kernel that extends over a few cells. We assumed that the core clock mechanisms of *Synechococcus* and *Anabaena* are similar (*Kushige et al., 2013*; *Schmelling et al., 2017*; *Garces et al., 2004*) and followed the interaction network depicted in *Figure 5A*. We further assumed that the rates of interconversion between KaiC phosphoforms for *Anabaena* are similar to those measured previously (*Rust et al., 2007*) in a reconstituted *in vitro* system consisting of KaiA, KaiB, and KaiC of *Synechococcus* (*Appendix 1—table 2*). This is supported by the fact that individual cells in both organisms exhibit oscillations with circadian periods (*Figure 2C*, *Figure 4B*) that in the case of *Synechococcus* are rather insensitive to changes in KaiC concentrations (*Chew et al., 2018*), constraining the values of these rates. Furthermore, KaiA of *Anabaena*, which is about two-thirds shorter than KaiA of *Synechococcus*, is similarly able to dimerize and enhance the phosphorylation of KaiC of other cyanobacteria *in vitro*, as well as elicit oscillations when its gene is transferred to *Synechococcus* cells, despite the evolutionary divergence between both organisms (*Uzumaki et al., 2004*). Individual cells can therefore display self-sustained oscillations only if the parameters $\gamma$ and [KaiA] take values inside the colored region of *Figure 5C*. These oscillations are characterized by a circadian period only within a narrow strip near the stability boundary. The fluorescence intensity is assumed to reflect the output of the clock, with a phase difference as shown previously (*Kushige et al., 2013*). We hence calculated the power spectrum of the fluorescence signal on every cell along the filament and averaged together the results. The experimentally computed power spectrum shows a clear peak, which is nicely interpolated by the theoretically predicted curve (*Figure 6C*) upon adjusting the two fitting parameters, $\gamma$ and [KaiA]. Again, the fitted values position the system (diamond in *Figure 5C*) outside the region of deterministic order, suggesting that demographic noise could trigger the observed oscillations. To test the robustness of the fit, we have implemented a bootstrap procedure (see Materials and methods) by perturbing each kinetic parameter with respect to the values reported in *Rust et al., 2007*. The imposed perturbation was about 10% for each individual kinetic parameter. The obtained best fit estimates for [KaiA] and $\gamma$ were found to display a degree of variability of approximately 20%, with reference to averaged values reported in *Figure 5C*. The bifurcation line that sets the separation between the deterministic limit cycle and the stationary stable fixed point became modulated depending on the set of assigned kinetic constants. Each pair of fitted $\gamma$ and [KaiA] falls by definition in the domain where the deterministic oscillations are impeded and the stochastic finite size corrections drive the emergence of the resonant quasi-cycles.

In *Anabaena*, the oscillations displayed by different cells along a filament are synchronized, most likely by cell-cell communication. To support further the notion of clock coupling, we studied the spatial coherence of noisy oscillations along an *Anabaena* filament. We assume that $P_{pecB-gfp}$

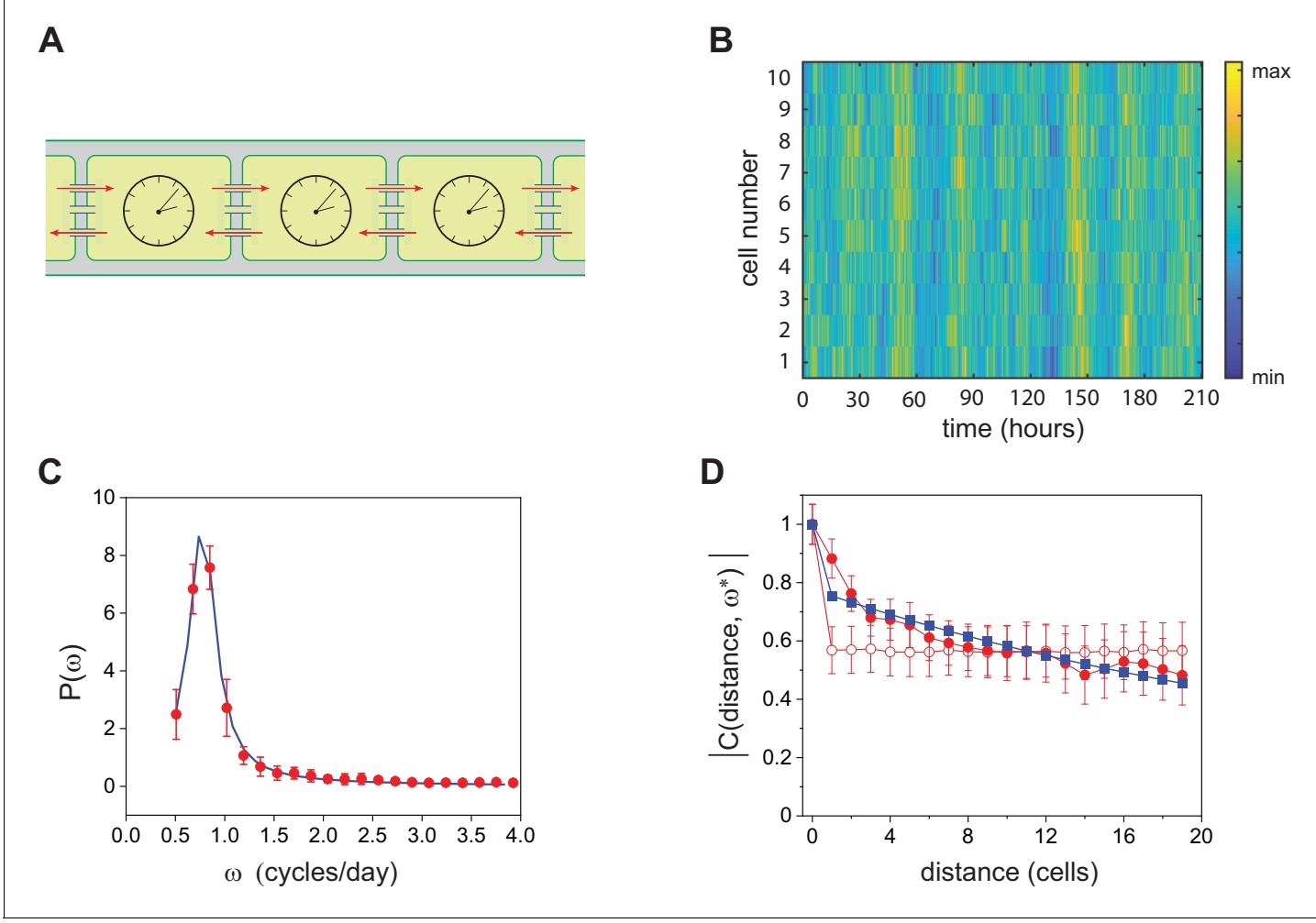

**Figure 6.** Stochastic model for circadian oscillations in *Anabaena*. (**A**) Schematic representation of the *Anabaena* filament showing coupling of circadian clocks via cell-cell communication (red arrows). (**B**) Gillespie simulations of quasi-cycles of T-KaiC in a continuous stretch of 10 cells along a filament. The reaction parameters correspond to the diamond plotted in *Figure 5C*. The total amount of KaiC phosphoforms was set to 5000, and the number of steps of the Gillespie algorithm was $1.6 \times 10^7$. (**C**) Average power spectrum of single-cell fluorescence intensity along filaments (red symbols) fit to the data with the prediction from the stochastic model (blue line). The best fit values correspond to the diamond shown in *Figure 5C*. (**D**) Complex coherence function measuring the correlation expression from $P_{pecB-gfp}$ in 35 cell segments at the frequency of temporal oscillations. Red full circles correspond to experimental data, blue squares represent the fit to the experimental data with the prediction of the stochastic model, and empty red circles represent the coherence function of the experimental data in which cells have been reshuffled. Lines between symbols are a guide to the eye. The fit was carried out by adjusting two parameters, the strength of the imposed spatial coupling and the characteristic scale of the exponential kernel, see Appendix 1. Remarkably, the range of the interaction as obtained from the fit is compatible with that estimated from the spatial autocorrelation depicted in *Figure 2D*. The intercell coupling was obtained from the fit in (**C**).

expression reports faithfully the output of the clock, albeit with a delay due to the transduction of KaiC's phosphorylation state up to activation of the promoter, probably by phosphorylated RpaA. To study spatial coherence, we calculated the complex coherence function (*Marple, 1987*), whose magnitude measures the degree of correlation between cells within a filament at different distances. As shown in *Figure 6D*, the coherence at the characteristic frequency of oscillations was a monotonically decaying function of distance, an observation that points to the existence of non-local interactions by cell-cell communication along the filament. As complementary evidence, we notice that the magnitude of the coherence function took a constant value when spatial correlations were broken by randomly reshuffling the position of the cells along the filament. Remarkably, the prediction of the stochastic model captures the experimental behavior, as shown in *Figure 6C*. The range of interaction of the exponential kernel is self-consistently quantified by the fitting strategy and yields a

result that agrees with that obtained from the spatial autocorrelation analysis (*Figure 2D*). More specifically, this conclusion is consistent with the reduction of spatial autocorrelation in the expression of a clock-regulated gene when either cell-cell communication is perturbed or clock genes are deleted, and with the notion that inheritance following cell division alone cannot account for all the spatial variation of expression along filaments (*Figure 2D*). In *Figure 6C*, quasi-cycles recorded on different cells of the stochastic *Anabaena* model are depicted showing a remarkable degree of synchronization. In Appendix 1, the analysis is extended so as to account for both a constant and a decaying power-law kernel of interaction.

## Discussion

*Anabaena*, a model organism in which each cell has a well-defined number of neighbors with which it communicates, has enabled us to study one-dimensional arrays of circadian clocks in a multicellular organism in space and time by interrogating each cell individually. Remarkably, our experiments using a fluorescence reporter fused to the N-terminus of a clock-controlled protein show that filaments display large-amplitude oscillations, consistently with previous studies carried out in bulk cultures (*Kushige et al., 2013*). These oscillations, which run freely under constant light conditions and are therefore autonomous, are characterized by high spatial coherence and synchrony. In addition to inheritance following cell division as in unicellular *Synechococcus* (*Amdaoud et al., 2007*), the behaviors of both the spatial autocorrelation (*Figure 2D*) and the complex coherence function (*Figure 6D*) represent strong evidence that these two characteristics result from local coupling between clocks of neighboring cells due to cell-cell communication via septal junctions. It is unlikely that this coupling results from the direct cell-to-cell transfer of KaiA, KaiB, or KaiC clock components as septal junctions in *Anabaena* are known to serve as conduits of only small molecules including metabolites, nutrients, and small peptides (e.g., PatS-derived peptides involved in lateral inhibition during developmental pattern formation, mediated by SepJ; *Corrales-Guerrero et al., 2015*). We postulate that metabolic factors such as sugars, which are transferred between cells via FraC and FraD proteins (*Mullineaux et al., 2008*; *Nürnberg et al., 2015*), may affect the behavior of neighboring clocks, yielding an effective long-ranged coupling between clock inputs (e.g., redox state, glycogen abundance, and ATP/ADP ratio) across the filament (*Cohen and Golden, 2015*; *Golden, 2020*). This is consistent with the shorter decay of the spatial autocorrelation function of $P_{pecB-gfp}$ fluorescence intensity, and with the reduced value of the synchronization index, measured in filaments of a strain in which the transport of sugars and/or other coupling factors is impaired ($\Delta sepJ/\Delta fraCD$). Of note, the value of the synchronization index in this strain is similar to that obtained for cells from different filaments. Thus, we have found that cells within a filament behave coordinately, whereas different filaments appear to be in different oscillatory phases. Nonetheless, coherent oscillations at the whole culture level were observed after light/darkness training of the cultures, showing that input signals can coordinate the circadian rhythm in cells from different filaments.

The high synchrony and spatial coherence observed in our experiments further suggest that circadian rhythms in *Anabaena* are not centrally coordinated as in higher multicellular organisms (*Bell-Pedersen et al., 2005*). In mammals, for instance, clocks in peripheral tissues are centrally coordinated from a central pacemaker in the brain, the suprachiasmatic nucleus (*Reppert and Weaver, 2002*). In plants such as Arabidopsis, recent studies revealed the existence of waves of gene expression across the whole plant and the response of cells to positional information (*Gould et al., 2018*), consistent with weak coupling between cells and supporting a more decentralized model of clock coordination in plants (*Endo, 2016*). High synchrony and spatial coherence may be viewed as a necessary adaptation following the transition from a unicellular to a multicellular lifestyle (*Masuda et al., 2017*), while preserving key architectural features, gating of the cell cycle by the circadian clock and robust response to stresses, supporting the notion that a filament represents the organismic unit in *Anabaena*.

Our single-cell measurements of gene expression from $P_{pecB-gfp}$ indicated that the cell-cell variability along filaments as measured by $CV^2$ is oscillating, achieving maxima approximately half-way during periods of increase of the fluorescence intensity. The phase difference between the oscillation in fluorescence intensity and its cell-cell variability is consistent with the phase of *rpaA* transcriptional oscillations observed in our RT-qPCR measurements. The oscillatory nature of $CV^2$ in $P_{pecB-gfp}$

expression may be due to the relay of the signal of the core clock by the transcriptional oscillations of a transcription factor, here RpaA (*Heltberg et al., 2019*).

Our experiments, carried out under constant light conditions, that is, non-varying external cues, provide clear evidence of gating of the cell cycle by the circadian clock, as for *Synechococcus* and many other organisms (*Mori, 2009*). In *Synechococcus*, cell doubling times vary considerably with light intensity (*Teng et al., 2013*), and the cell cycle has no effect on the circadian clock, irrespective of the cell cycle rate (*Mori et al., 1996*; *Mori and Johnson, 2001*). Furthermore, as for *Synechococcus* (*Mihalcescu et al., 2004*; *Mori et al., 1996*; *Yang et al., 2010*), clock phase is faithfully inherited by any two daughters following cell division as illustrated in *Figure 2C*. In fact, the post-transcriptional design of cyanobacterial circadian circuits has been suggested to provide insulation from effects due to variable cell division rates (*Paijmans et al., 2016*). Cell doubling times in *Anabaena* can also vary significantly with light intensity (*Zhao et al., 2007*). The mechanism of how the circadian clock controls the timing of cell division has been studied in *Synechococcus* as well as in *Synechocystis*, and it has been suggested that phosphorylated RpaA regulates the bacterial tubulin-analog FtsZ, inhibiting the formation of the cytokinetic ring (*Dong et al., 2010*; *Kizawa and Osanai, 2020*). While the precise genetic and biochemical differences between the circadian clocks of *Anabaena* and *Synechococcus* remain to be elucidated, the presence of homologs of core clock components, output coupling factors such as RpaA (*Schmelling et al., 2017*), and cell division promoters such as FtsZ in *Anabaena* (*Corrales-Guerrero et al., 2018*) suggests that the mechanisms for gating in these two organisms may be similar. Note that *ftsZ* has an upstream putative RpaA binding site motif (see *Figure 3—figure supplement 2*). Interestingly, nitrogen-fixing cells – heterocysts – that are formed under nitrogen deprivation in *Anabaena* lose the ability to divide, and yet, the heterocyst-enriched fraction in bulk experiments has been shown to exhibit clear circadian oscillations (*Kushige et al., 2013*). This supports the notion that cell division does not affect the circadian clock in *Anabaena* as in *Synechococcus*.

The RT-qPCR results and their analysis provide conclusive and quantitative evidence for circadian oscillations in the expression of *kai* genes in *Anabaena*. The calculation of the synchronization index *R* for different pairs of *kai* genes yields large and similar values, suggesting that the expression of the three genes is highly coordinated, consistent with their possible expression as an operon. Furthermore, a persistent homology analysis of the time series of the three genes yields clear evidence of oscillatory behavior, consistent with a circadian period. In agreement with microarray measurements (*Kushige et al., 2013*), the oscillations we detect are of small amplitude, similar to those of *Synechocystis* (*Kucho et al., 2005*; *Beck et al., 2014*), but unlike those of *Synechococcus*. Given that the oscillations in the expression of *kai* genes are small, but that of *pecB* and many other targets are large (*Kushige et al., 2013*), we reasoned that a possible way of transducing the core clock signal to clock-controlled genes may be furnished by the downstream transcription factor RpaA. The *kai*-dependent transcriptional oscillations of *rpaA* we observed in *Anabaena* suggest an additional level of regulation of clock-controlled genes, which differs from the post-translational mechanism observed in *Synechococcus*. In particular, the oscillations in *kai* genes are consistent with the existence of a transcription-translation feedback loop via RpaA (*Markson et al., 2013*). Indeed, a well-defined candidate binding site motif of RpaA is located upstream of the *kai* genes (*Figure 3—figure supplement 2*). Additional candidate binding site motifs of RpaA are found upstream of the *rpaA* locus itself as well as the *pecBACEF* operon. The mechanism behind the large-amplitude oscillatory behavior of *rpaA* transcription we observed remains to be elucidated.

In order to describe theoretically circadian oscillations in individual *Anabaena* filaments, we started by incorporating demographic noise into a deterministic model of a single clock that includes only the *kai* genes as a core, such as the one describing oscillations in *Synechococcus*, and then extended the model to one describing a one-dimensional array of coupled noisy clocks as in *Anabaena*. The stochastic models capture well the peaks of finite width in the power spectra of fluctuations observed in our experiments (*Figure 5B*, *Figure 6C*). Importantly, the parameters we obtain from the fits to the experimental power spectra lie well outside the region in parameter space where deterministic oscillations occur, and in particular, the smaller region corresponding to oscillations characterized by circadian periods. This strongly suggests that the oscillatory behavior may correspond to noise-seeded oscillations as observed in other systems (*McKane and Newman, 2005*), without the need of training by light-dark cycles. The noise-seeded enlargement of the range of biological parameters over which circadian oscillatory behavior can be observed may confer robustness

to clocks, a decisive biological advantage. Robustness may enable proper circadian clock function of *Anabaena* under a variety of stresses, including those involved in metabolic rewiring and the ensuing differentiation in response to combined nitrogen deprivation (*Herrero et al., 2016*; *Di Patti et al., 2018*). In this context, note that *Anabaena* filaments display circadian oscillations under nitrogen-deprived conditions, as shown previously (*Kushige et al., 2013*). Lastly, the model also reproduces the spatial coherence of oscillations in filaments, providing independent evidence of clock coupling through cell-cell communication.

In spite of the fact that Kai protein copy numbers run in the thousands in *Synechococcus*, phase fluctuations between cells are readily visible both in our experiments and in previous ones (*Chew et al., 2018*; *Chabot et al., 2007*). Stochastic modeling indicates that these numbers may be needed to compensate for noise amplification introduced by the post-translational feedback loop provided by KaiA, and reducing the numbers of Kai proteins leads to lower clock precision (*Chew et al., 2018*). We point out that demographic noise may be more significant than the copy numbers of Kai proteins may suggest: the functional form of KaiC is hexameric, which further partitions into its different phosphoforms (*Chew et al., 2018*). In addition, it has been shown that KaiB spends most of its time in an inactive tetrameric state, preventing its interaction with other Kai proteins, and that KaiA also oscillates between active and inactive states by binding to KaiB (*Tseng et al., 2017*). Together, these regulatory contributions might reduce active KaiC-effective numbers, thereby increasing fluctuations.

While the copy numbers of Kai proteins in *Anabaena* have not been measured, it is likely that they are significantly smaller than in *Synechococcus*, given the low transcriptional levels observed in our RT-qPCR experiments, as well as in DNA microarrays (*Kushige et al., 2013*). The expression of *kai* genes in *Synechocystis* is weak (*Kucho et al., 2005*), and protein copy numbers are correspondingly small (*Zavřel et al., 2019*). Thus, demographic noise effects may be indeed at least as important in *Anabaena* as they are for *Synechococcus*. One may hypothesize that cell-cell communication and the resulting coupling of clocks compensate for the smaller number of Kai proteins, setting the high synchrony and spatial coherence we observe in *Anabaena*. The picture that emerges may be more general and applicable to other multicellular organisms (*Gould et al., 2018*).

The transition from unicellular to multicellular organisms demanded coordination between physiological processes in different cells in order to enhance organismal fitness and adaptation to stresses. This is true in particular of circadian clocks and the genes they regulate. By analyzing filaments at the individual cell level, we found concrete evidence supporting coordination mediated by cell-cell communication, allowing clocks in different cells to be coupled. Furthermore, the high synchrony and spatial coherence of cell clocks along filaments observed in the experiments suggest that cell-cell communication contributes significantly to these two properties. Our theoretical model of single core clocks and arrays thereof shows that far from being detrimental demographic noise may seed oscillations that can be synchronized by clock coupling. This provides a robust description of circadian oscillations in a multicellular organism such as *Anabaena*.

## Materials and methods

**Key resources table**

| Reagent type (species) or resource | Designation | Source or reference | Identifiers | Additional information |
|---|---|---|---|---|
| Strain, strain background (*Anabaena*) | $P_{pecB-gfp}$, WT | This paper | | *Anabaena* PCC 7120 WT, bearing a *pecB* promoter fusion to *gfp* |
| Strain, strain background (*Anabaena*) | $P_{pecB-gfp}$, $\Delta kaiABC$ | This paper | | *Anabaena* PCC 7120 deletion mutant of the *kaiABC* genes, bearing a *pecB* promoter fusion to *gfp* |

*Continued on next page*

Continued

| Reagent type (species) or resource | Designation | Source or reference | Identifiers | Additional information |
|---|---|---|---|---|
| Strain, strain background (*Anabaena*) | P*hetR−gfp* | doi: 10.1371/journal.pgen.1005031 | CSL64 | *Anabaena* PCC 7120 WT, bearing a *hetR* promoter fusion to *gfp* |
| Strain, strain background (*Anabaena*) | P*pecB−gfp*, Δ*sepJ*/Δ*fraC*/Δ*fraD* | This paper | | *Anabaena* PCC 7120 deletion mutant of the *sepJ*, *fraC*, *fraD* genes (CSVM141), bearing a *pecB* promoter fusion to *gfp* |
| Strain, strain background (*Synechococcus elongatus*) | YFP-SsrA | This paper | PCC 7942 | *Synechococcus elongatus* PCC 7942 (wild-type) expressing YFP- SsrA |
| Recombinant DNA reagent | EB2316 (plasmid) | Addgene plasmid | 87753 | http://n2t.net/addgene: 87753 |
| Recombinant DNA reagent | pSpark (plasmid) | Canvax | C0001 | https://lifescience. canvaxbiotech.com/ wpcontent/uploads/sites/ 2/2015/ 08pSpark-DNA-Cloning. pdf |
| Recombinant DNA reagent | pCSRO *sacB−* containing cloning vector | doi: 10.1128/JB.00181-13 | | |
| Commercial assay or kit | Fast SYBR Green Master Mix | Applied Biosystems | 4385612 | |
| Commercial assay or kit | QuantiTect Reverse Transcription kit | QIAGEN | 205311 | |

## Strains

Strains bearing a chromosomally encoded P*pecB−gfp*, were obtained by conjugation with the following backgrounds: WT *Anabaena* sp. (also known as *Nostoc* sp.) strain PCC 7120; Δ*sepJ*/Δ*fraCD*, strain CSVM141 in which *sepJ*, *fraC*, and *fraD* were deleted (*Nürnberg et al., 2015*); Δ*kaiABC* in which the *kaiABC* genes were deleted (for details, see Appendix 1 – Supplemental methods), as recipients. Strain CSL64 bearing a chromosomally encoded P*hetR−gfp* transcriptional fusion in a WT background has been reported previously (*Corrales-Guerrero et al., 2015*). The *S. elongatus* strain containing the gene encoding a YFP-SsrA reporter whose expression is driven by the *kaiBC* promoter at neutral site II was constructed by transforming *Synechococcus* sp. PCC 7942 with plasmid EB2316 (Addgene plasmid # 87753; http://n2t.net/addgene:87753; RRID: Addgene 87753).

## Culture conditions

Strains and derived strains were grown photoautotrophically in BG11 medium containing NaNO$_3$, supplemented with 20 mM HEPES (pH 7.5) with shaking at 180 rpm, at 30℃, as described previously (*Corrales-Guerrero et al., 2013*; *Di Patti et al., 2018*). Growth took place under constant illumination (10 µmol of photons [spectrum centered at 450 nm]) from a cool-white LED array. When required, streptomycin sulfate (Sm) and spectinomycin dihydrochloride pentahydrate (Sp) were added to the media at final concentrations of 2 µg/mL for liquid and 5 µg/mL for solid media (1% Difco agar). The densities of the cultures were adjusted so as to have a chlorophyll content of 2–4 µg/mL 24 hr prior to the experiment, following published procedures (*Di Patti et al., 2018*). For time-lapse measurements, filaments in cultures were harvested and concentrated 50-fold. *Synechococcus* cultures were grown as above, and when required, 7.5 µg chloramphenicol (Cm) was added.

## Samples for time-lapse microscopy

Strains were grown as described previously (*Di Patti et al., 2018*). When required, antibiotics, Sm and Sp were added to the media at final concentrations of 2 µg/mL for liquid and 5 µg/mL for solid media. The densities of the cultures, grown under an external LED array (15 µmol for about 5 days), were adjusted so as to have a chlorophyll *a* content of 2–4 µg/mL, 24 hr prior to the experiment following published procedures (*Di Patti et al., 2018*). For time-lapse, single-cell measurements of *Anabaena*, 5 µL of culture concentrated 100-fold were pipetted onto an agarose low-melting gel pad (1.5%) in BG11 medium containing $NaNO_3$ and 10 mM $NaHCO_3$, which was placed on a microscope slide. The pad with the cells was then covered with a #0 mm coverslip and then placed on the microscope at 30°C. The cells grew under light from both an external LED array (15 µmol) and tungsten halogen light (10 µmol, 3000K color). Under these illumination conditions, the doubling time of cells is similar to that in bulk cultures (*Di Patti et al., 2018*). The change in illumination conditions when transferring cells from bulk cultures to the microscope results in high synchronization within filaments. Images of about 10 different fields of view were taken every 30 min on a Nikon Eclipse Ti-E microscope controlled by the NIS-Elements software using a 60 N.A. 1.40 oil immersion phase contrast objective lens (Nikon plan-apochromat 60 1.40) and an Andor iXon X3 EMCCD camera. Focus was maintained throughout the experiment using a Perfect Focus System (Nikon). All the filters used are from Chroma. The filters used were ET480/40X for excitation, T510 as dichroic mirror, ET535/50M for emission (GFP set), ET500/20x for excitation, T515lp as dichroic mirror, and ET535/30m for emission (EYFP set), and ET430/24x for excitation, 505dcxt as dichroic mirror, and HQ600lp for emission (chlorophyll set). Samples were excited with a pE-2 fluorescence LED illumination system (CoolLED).

For measurements of *Synechococcus*, the cultures were grown as above to a OD750 of 0.3–0.4. Samples were then entrained by two light-dark cycles (12 hr–12 hr) before measurements commenced. The cultures were then diluted to a OD750 of 0.1 using BG11 medium, and 2 µL of the culture was pipetted onto a glass-bottom culture dish. A patterned pad prepared as above but solidified on an optical grating (*Hadizadeh Yazdi et al., 2012*) was placed atop the cell suspension. Then, 1.5% agarose melted in BG11, cooled to 37°C, was poured on top of the well to cover the pad. After solidifying this last layer, 5 mL of BG11 was added to the culture dishes to maintain the moisture level of the agarose pad during the experiment. This device was placed in the microscope, and time-lapse measurements were carried out as for *Anabaena*.

## RT-qPCR measurements

For RT-qPCR measurements, strains were grown under the same conditions as described above. When the cultures reached the beginning of their exponential phase, about 1.8 µg/mL of chlorophyll *a* content, they were entrained by two light-dark cycles (12 hr–12 hr) and then grown under constant light. Then, measurements were started 24 hr after, and total RNA was extracted from the PCC 7120 and Δ*kaiABC* cultures in two biological replicates, every 4 hr. For each sample, 20 mL of cells were collected by filtration and washed with buffer TE50 (50 mM Tris-HCl at pH 8.0, 100 mM EDTA). Cells were resuspended in 2 mL TE50 buffer, and then they were centrifuged at 11,500 *g* for 2 min at 4°C. Supernatant was then removed and cell pellets were flash-frozen in nitrogen liquid for storage at 80°C. Total RNA was isolated by using hot phenol as described (*Mohamed and Jansson, 1989*) with modifications. All samples were treated with DNase I, and RNA quantity and purity were assessed with a NanoDrop One spectrophotometer as well as by agarose gel electrophoresis.

For cDNA synthesis, 600 ng of total RNA of each sample were used for reverse transcription with the QuantiTect Reverse Transcription kit (QIAGEN). Then, PCR amplification of 17.4 ng of each cDNA was carried out using Fast SYBR Green Master Mix (Applied Biosystems) that includes an internal reference based on the ROX dye and specific primers (*Appendix 1—table 2*). The amplification protocol was one cycle at 95°C for 10 min, 40 cycles of 95°C for 15 s, and 60°C for 60 s. RT-qPCR was carried out using an Applied Biosystems StepOnePlus instrument equipped with the StepOne Software v2.3. After the amplification was completed, a melting point calculation protocol was carried out in order to check that only the correct product was amplified in each reaction. Reactions were run in triplicate in 3–4 independent experiments from two biological replicates. The transcript levels of *kaiA* (*alr2884*), *kaiB* (*alr2885*), *kaiC* (*alr2886*), *pecB* (*alr0523*), and *rpaA* (*all0129*) were

normalized to the transcript levels of the housekeeping genes *rnpB* and *all5167* (*ispD*) to obtain the $\Delta$ct value. Relative gene expression was calculated as $2^{-\Delta\Delta ct}$ (*Pfaffl, 2001*).

## Image segmentation

All image processing and data analysis were carried out using MATLAB (MathWorks). Filament and individual cell recognition was performed on phase contrast images using an algorithm developed in our laboratory. The program's segmentation was checked in all experiments and corrected manually for errors in recognition. The total fluorescence from GFP (for *Anabaena*) and chlorophyll (autofluorescence) channels of each cell, as well as the cell area, was obtained as output for further statistical analysis.

## Analysis of synchronization and spatial correlations along filaments

Synchronization was measured by the order parameter proposed by *Garcia-Ojalvo et al., 2004*:

$$R = \frac{\langle \overline{\mu^2} \rangle - \langle \overline{\mu} \rangle^2}{\overline{\langle f_i^2 \rangle - \langle f_i \rangle^2}} \tag{1}$$

where $\langle \cdot \rangle$ denotes a time average, $\overline{\phantom{.}}$ indicates an average over all cells, and $\mu$ denotes the average of the fluorescence intensity of each cell $f_i$. Hence, $R$ is defined as the ratio of the standard deviation of $\mu(t)$ to the standard deviation of $f_i$, averaged over all cells. For the measurement of synchronization within the same filament, groups of 8–11 cells were chosen, whether separated or contiguous (sharing a common ancestor as determined from a lineage analysis). For evaluation of inter-filament synchronization, one cell per filament was chosen randomly in different fields of view. $R$ was then calculated, and this procedure was repeated for different choices of cells, at least three times for each experiment. All the evaluations of $R$ were carried out over a full period of oscillation in either one of the first two oscillations, except for the $\Delta kaiABC$ background, for which $R$ was calculated for an interval of 24 hr, during which other strains display the first full oscillation. The final result comprises the mean of at least three independent repeats in at least two independent experiments. Errors in the quoted values of $R$ therefore represent standard errors (SEM). Statistical analyses were performed in MATLAB using Mann–Whitney U-test.

The spatial autocorrelation function of fluorescence intensities along filaments was calculated using the *autocorr* MATLAB command with 30 lags from at least 25 filaments of 50 cells each and at least two independent experiments. For WT and $\Delta sepJ/\Delta fraCD$ backgrounds, autocorrelation functions of individual filaments were calculated at maxima and minima of circadian oscillations and the results were averaged. For filaments of the $\Delta kaiABC$ background, autocorrelation functions were calculated at about 30 and 50 hr, corresponding to the first minimum and maximum of oscillations observed in filaments of WT background. The spatial autocorrelation function was calculated from the formula

$$g_k = \frac{c_k}{c_0} \tag{2}$$

where

$$c_k = \frac{1}{N} \sum_{i=1}^{N-k} (f_i - \bar{f})(f_{i+k} - \bar{f}) \tag{3}$$

where $f_i$ denotes the fluorescence intensity in cell $i$ in a filament of $N$ cells.

## Robustness of *Anabaena* dataset fit to the *Synechococcus* parameters

Each kinetic parameter ($k$) was randomly selected from a Gaussian distribution centered at the nominal value ($\bar{k}$) estimated by *Rust et al., 2007* using the *normrnd* MATLAB function. The standard deviation is assigned as $\sigma = \bar{k}/10$. For every complete set of (randomly selected) kinetic constants, we proceeded with the fit of $\gamma$ and [KaiA] to interpolate the experimental power spectrum. Each pair of fitted values was stored to eventually compute averaged estimates, together with the error, as quantified by the associated standard deviation. By averaging over 200 independent realizations of the implemented procedure yields $\gamma = 7.2 \pm 1.4$ and [KaiA] $= 1.3 \pm 0.24$ (mean $\pm$ SD).

## Acknowledgements

We thank M Feingold, M Camarena, T Unger, JE Frías, N Rosenthal, and O Vardi for help, and M Rust for strains. This study was supported by the Minerva Foundation (JS); grant numbers BFU2017-88202-P (EF) and BUF2016-77097-P (AH) from Plan Nacional de Investigación, Spain, co-financed by the European Regional Development Fund; Fondazione Ente Cassa di Risparmio, Florence, Italy (DF, FDP, VB); Project EXPLICS granted by the Italian Ministry of Foreign Affairs and International Cooperation (FDP).

## Additional information

### Funding

| Funder | Grant reference number | Author |
|---|---|---|
| Minerva Foundation | | Joel Stavans |
| Fondazione Ente Cassa di Risparmio di Firenze | | Duccio Fanelli |
| European Regional Development Fund | BUF2016-77097-P | Antonia Herrero |
| European Regional Development Fund | BFU2017-88202-P | Enrique Flores |
| Ministry of Foreign Affairs | EXPLICS | Francesca Di Patti |

The funders had no role in study design, data collection and interpretation, or the decision to submit the work for publication.

### Author contributions

Rinat Arbel-Goren, Conceptualization, Resources, Data curation, Formal analysis, Validation, Investigation, Visualization, Methodology, Writing - original draft, Project administration, Writing - review and editing; Valentina Buonfiglio, Software, Formal analysis; Francesca Di Patti, Conceptualization, Resources, Software, Formal analysis, Validation, Investigation, Visualization, Methodology, Writing - review and editing; Sergio Camargo, Resources, Formal analysis, Investigation, Visualization; Anna Zhitnitsky, Formal analysis; Ana Valladares, Resources, Validation, Investigation; Enrique Flores, Conceptualization, Resources, Supervision, Funding acquisition, Methodology, Writing - review and editing; Antonia Herrero, Conceptualization, Resources, Supervision, Funding acquisition, Writing - review and editing; Duccio Fanelli, Conceptualization, Resources, Formal analysis, Supervision, Funding acquisition, Visualization, Methodology, Project administration, Writing - review and editing; Joel Stavans, Conceptualization, Resources, Data curation, Software, Formal analysis, Supervision, Funding acquisition, Validation, Investigation, Visualization, Methodology, Writing - original draft, Project administration, Writing - review and editing

### Author ORCIDs

Rinat Arbel-Goren ORCID https://orcid.org/0000-0002-7253-2036
Valentina Buonfiglio ORCID https://orcid.org/0000-0003-1528-2955
Francesca Di Patti ORCID https://orcid.org/0000-0003-1288-0079
Sergio Camargo ORCID https://orcid.org/0000-0002-5688-3215
Anna Zhitnitsky ORCID https://orcid.org/0000-0001-9598-4812
Ana Valladares ORCID https://orcid.org/0000-0002-6683-1355
Enrique Flores ORCID https://orcid.org/0000-0001-7605-7343
Antonia Herrero ORCID https://orcid.org/0000-0003-1071-6590
Duccio Fanelli ORCID https://orcid.org/0000-0001-8545-9424
Joel Stavans ORCID https://orcid.org/0000-0003-0396-7797

### Decision letter and Author response

Decision letter https://doi.org/10.7554/eLife.64348.sa1

Author response https://doi.org/10.7554/eLife.64348.sa2

# Additional files

## Supplementary files
- Transparent reporting form

## Data availability

Source data files, Video 1 and Key resources table have been deposited in Dryad (https://doi.org/10.5061/dryad.sxksn031n).

The following dataset was generated:

| Author(s) | Year | Dataset title | Dataset URL | Database and Identifier |
|---|---|---|---|---|
| Arbel-Goren R, Buonfiglio V, Patti FD, Camargo S, Zhitnitsky A, Valladares A, Flores E, Herrero A, Fanelli D, Stavans J | 2020 | Demographic noise seeds robust synchronized oscillations in the circadian clock of Anabaena | http://dx.doi.org/10.5061/dryad.sxksn031n | Dryad Digital Repository, 10.5061/dryad.sxksn031n |

The following previously published datasets were used:

| Author(s) | Year | Dataset title | Dataset URL | Database and Identifier |
|---|---|---|---|---|
| Kaneko T, Nakamura Y, Wolk PC, Kuritz T, Sasamoto S, Watanabe A, Iriguchi M, Ishikawa A, Kawashima K, Kimura T, Kishida Y, Kohara M, Matsumoto M, Matsuno A, Muraki A, Nakazaki N, Shimpo S, Sugimoto M, Takazawa M, Yamada M, Yasuda M, Tabata S | 2001 | Complete Genomic Sequence of the Filamentous Nitrogen-fixing Cyanobacterium *Anabaena* sp. Strain PCC 7120 | http://www.kazusa.or.jp/cyanobase/ | Accession numbers, AP003581 (nucleotide positions 1-348,050), AP003582 (348,001- 690,650), AP003583 (690,601-1,030,250), AP003584 (1,030,251-1,378,550), AP003585 (1,378,501-1,720,550), AP003586 (1,720,501-2,069,550), AP003587 (2,069,501- 2,413,050), AP003588 (2,413,001-2,747,520), AP003589 (2,747,471-3,089,350), AP003590 (3,089,301-3,422,800), AP003591 (3,422,751-3,770,150), AP003592 (3,770,101- 4,118,350), AP003593 (4,118,301-4,451,850), AP003594 (4,451,801-4,795,050), AP003595 (4,795,001-5,142,550), AP003596 (5,142,501-5,491,050), AP003597 (5,491,001- 5,833,850), AP003598 (5,833,801-6,176,600), and AP003599 (6,176,551-6,413) The sequence data analyzed in this study have been registered in DDBJ/GenBank/EMBL, 771 |
| Markson JS, Piechura JR, Puszynska AM, O'Shea EK | 2013 | Circadian Control of Global Gene Expression by the Cyanobacterial Master Regulator RpaA | https://www.ncbi.nlm.nih.gov/geo/query/acc.cgi?acc=GSE32317 | NCBI Gene Expression Omnibus, GSE50922 |
| Copeland A, Lucas S, Lapidus A, Barry K, Detter JC, Glavina T, Hammon N, Israni S, Pitluck S, Schmutz J, | 2014 | *Synechococcus elongatus* PCC 7942, complete genome | https://www.ncbi.nlm.nih.gov/nuccore/CP000100 | NCBI GenBank, CP000100.1 |

Larimer F, Land M, Kyrpides N, Lykidis A, Golden S, Richardson P

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

## Appendix 1

### Persistent homology analysis of periodic behavior

To apply persistent homology methods for the detection of oscillatory behavior in the noisy low-amplitude time series of *kaiA*, *kaiB,* and *kaiC*, a two-dimensional representation of each time series was constructed by a time-delayed embedding, that is, a plot of the signal versus itself at two different time points separated by a delay $\tau$ (see *Figure 3—figure supplement 1*). To overcome the sparsity of data (12 time points) when plotting the time-delayed embedding, each time series was first padded by intermediate points determined by linear interpolation. The delay $\tau$ was then evaluated as the first minimum of the mutual information of the series, as determined by a fifth degree polynomial fit, following previously published procedures (*Fraser and Swinney, 1986*). The existence of an underlying oscillation in the data was assessed by using a MATLAB algorithm based on the JavaPlex package (*Tausz et al., 2014*), which yields as output a barcode diagram in which each bar represents the range of scales over which a cycle persists as observed in the phase portrait, regarded as a simplicial complex. Note that for a periodic signal the delay $\tau$ corresponds to a quarter of the signal's period (*Kennedy et al., 2018*).

### Stochastic model

We model the *Anabaena* filament as a one-dimensional chain, where each node represents a cell with its individual circadian clock. We use the letter $\alpha$ to label the node and denote by $\Omega$ the length of the chain, so that $\alpha = 1, \ldots, \Omega$ (see *Appendix 1—figure 1*). As explained in the main text, the coupling between clocks that result from metabolic factors yields a long-range interaction between clocks along the filament. To account for this crucial phenomenon, we schematize the chain as a network and, therefore, introduce the adjacency matrix $W$ whose elements $W_{\alpha\beta}$ are 1 if nodes $\alpha$ and $\beta$ are connected, 0 otherwise. The connectivity of each node is assumed to be constant for each node and is denoted by $k_\alpha$.

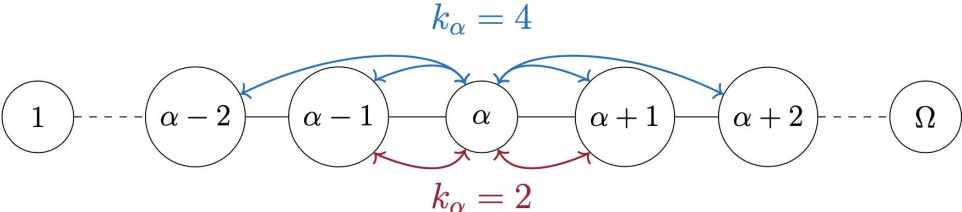

**Appendix 1—figure 1.** Schematic representation of an *Anabaena* filament. The parameter $k_\alpha$ measures the connectivity of each node.

Referring back to *Figure 3A*, we consider, in each cell, the cyclic interconversion between phosphoforms of KaiC, T, S, D (ST), and U encapsulated in the following chemical equations:

$$
\begin{array}{llll}
U^\alpha \xrightarrow{k_{UT}} T^\alpha & T^\alpha \xrightarrow{k_{TU}} U^\alpha & D^\alpha \xrightarrow{k_{DT}} T^\alpha & S^\alpha \xrightarrow{k_{SU}} U^\alpha \\
U^\alpha \xrightarrow{k_{US}} S^\alpha & T^\alpha \xrightarrow{k_{TD}} D^\alpha & D^\alpha \xrightarrow{k_{DS}} S^\alpha & S^\alpha \xrightarrow{k_{SD}} D^\alpha
\end{array}
\tag{4}
$$

where the superscript $\alpha$ indicates that the species belongs to the cell $\alpha$. The interconversion rates between phosphoforms are given by

$$
k_{XY}\left(\frac{n_S^\alpha}{N}\right) = k_{XY}^0 + k_{XY}^A f\left(\frac{n_S^\alpha}{N}\right) \quad \text{for}(X,Y) \in \{U, T, D, S\}
\tag{5}
$$

with

$$
f\left(\frac{n_S^\alpha}{N}\right) = \frac{\Gamma}{1 + \exp\left[\frac{\gamma}{[\text{KaiA}]}\left(\frac{n_S^\alpha [\text{KaiC}]}{N} - \frac{[\text{KaiA}]}{2}\right)\right]}
\tag{6}
$$

where [KaiA] (resp. [KaiC]) is the concentration of KaiA (resp. [KaiC]). The values of the rates are set as those of *Lambert et al., 2016* and are listed in *Appendix 1—table 1*. The variable $n_S^\alpha$ stands for

the total number of S molecules in cell $\alpha$ at time $t$, and, in the same way, we will use $n_X^\alpha$ to denote the total number of X molecules for $X = T, D, U$. The total number of KaiC phosphoforms remains constant in each cell of the filament, and we set this value equal to $N$. Note that KaiB does not appear explicitly in the equations as it is assumed that it is in excess at all times and it is further assumed that it interacts with S-KaiC instantaneously (*Lambert et al., 2016*). The typical sigmoidal shape of the nonlinear function (*Teng et al., 2013*) is plotted in *Figure 5—figure supplement 1* (blue dashed line), where we also plotted the corresponding function used in the paper by *Lambert et al., 2016* (red line).

**Appendix 1—table 1.** Parameters used in the simulations.

| | | | |
|---|---|---|---|
| $k_{UT}^0$ | 0 h$^{-1}$ | $k_{UT}^A$ | 0.479077 h$^{-1}$ |
| $k_{TD}^0$ | 0 h$^{-1}$ | $k_{TD}^A$ | 0.212923 h$^{-1}$ |
| $k_{SD}^0$ | 0 h$^{-1}$ | $k_{SD}^A$ | 0.505692 h$^{-1}$ |
| $k_{US}^0$ | 0 h$^{-1}$ | $k_{US}^A$ | 0.0532308 h$^{-1}$ |
| $k_{TU}^0$ | 0.21 h$^{-1}$ | $k_{TU}^A$ | 0.0798462 h$^{-1}$ |
| $k_{DT}^0$ | 0 h$^{-1}$ | $k_{DT}^A$ | 0.1730000 h$^{-1}$ |
| $k_{DS}^0$ | 0.31 h$^{-1}$ | $k_{DS}^A$ | −0.319885 h$^{-1}$ |
| $k_{SU}^0$ | 0.11 h$^{-1}$ | $k_{SU}^A$ | −0.133077 h$^{-1}$ |
| $k_{1/2}$ | 0.43 µM | [KaiC] | 3.4 µM |

All the values are from *Lambert et al., 2016*.

To model cell-cell communication, we postulate a constant kernel that extends over few cells, as depicted in *Appendix 1—figure 1*. This hypothesis will be relaxed later (see Accounting for a smooth kernel of interactions). More specifically, and despite the fact that there is no experimental evidence for intercellular transport of KaiC, we assume that coupling between clocks is metabolic and describe this effect through long-range interactions between molecules at different nodes. The parameter $k_\alpha$, as shown in *Appendix 1—figure 1*, counts the number of neighbors of any given cell. The chemical equations corresponding to the envisaged coupling read

$$T^\alpha \xrightarrow{\nu} T^\beta \qquad D^\alpha \xrightarrow{\nu} D^\beta \qquad S^\alpha \xrightarrow{\nu} S^\beta \tag{7}$$

where $\alpha$ and $\beta$ label connected nodes.

Since the total number of molecules within each cell of the filament remains constant, the variable $n_U^\alpha$ can be neglected because it can be recovered from the conservation law $N = n_U^\alpha + n_S^\alpha + n_T^\alpha + n_D^\alpha$. For this reason, at each time $t$ the state of the system is specified by the $3\Omega$-dimensional vector $\boldsymbol{n}(t)$

$$\boldsymbol{n}(t) = (n_T^1(t), n_D^1(t), n_S^1(t); \ldots; n_T^\Omega, n_D^\Omega, n_S^\Omega) = (\boldsymbol{n}^1(t), \ldots, \boldsymbol{n}^\Omega(t)) \tag{8}$$

The microscopic dynamics described by rules (*Equations 4 and 7*) is Markovian, and thus the configuration of the system only depends on that at the previous time. We introduce the transition rates $\mathbb{T}(\boldsymbol{n}'|\boldsymbol{n})$ from state $\boldsymbol{n}$ to a new state $\boldsymbol{n}'$ for all the chemical equations that constitute the stochastic model. To denote the new state $\boldsymbol{n}'$, we only explicitly write the components of the vector that change. Accordingly, the phosphorylation and dephosphorylation transition rates in each cell $\alpha$ read

$$\mathbb{T}(n_i^\alpha + 1|\boldsymbol{n}) = \frac{n_U^\alpha}{N} k_{Ui}^{(\alpha)} \qquad i \in \{\mathrm{T, S}\} \tag{9}$$

$$\mathbb{T}(n_i^\alpha - 1|\boldsymbol{n}) = \frac{n_i^\alpha}{N} k_{iU}^{(\alpha)} \qquad i \in \{\mathrm{T, S}\} \tag{10}$$

while the interconversion transitions are

$$\mathbb{T}(n_i^\alpha - 1, n_j^\alpha + 1|\boldsymbol{n}) = \frac{n_i^\alpha}{N} k_{ij}^{(\alpha)} \qquad (i, j) \in \{(\mathrm{T, D}), (\mathrm{D, T}), (\mathrm{D, S}), (\mathrm{S, D})\} \tag{11}$$

To account for cell-cell communication, we also add the transitions for long-range interaction between two connected cells $\alpha$ and $\beta$:

$$\mathbb{T}(n_i^\alpha - 1, n_i^\beta + 1|\boldsymbol{n}) = \frac{n_i^\alpha}{N}\nu \qquad i \in \{\mathrm{T}, \mathrm{D}, \mathrm{S}\} \tag{12}$$

The dynamics is fully described in terms of the following master equation:

$$\frac{\partial P(\boldsymbol{n},t)}{\partial t} = \sum_{\boldsymbol{n}' \neq \boldsymbol{n}} [\mathbb{T}(\boldsymbol{n}|\boldsymbol{n}')P(\boldsymbol{n}',t) - \mathbb{T}(\boldsymbol{n}'|\boldsymbol{n})P(\boldsymbol{n},t)] \tag{13}$$

whose explicit version reads

$$
\begin{aligned}
\frac{\partial P(\boldsymbol{n},t)}{\partial t} =& \sum_{\alpha=1}^{\Omega} \{\mathbb{T}(\boldsymbol{n}|n_T^\alpha - 1)P(n_T^\alpha - 1;t) - \mathbb{T}(n_T^\alpha - 1|\boldsymbol{n})P(\boldsymbol{n},t) + \\
&+ \mathbb{T}(\boldsymbol{n}|n_T^\alpha - 1, n_D^\alpha + 1)P(n_T^\alpha - 1, n_D^\alpha + 1;t) - \mathbb{T}(n_T^\alpha - 1, n_D^\alpha + 1|\boldsymbol{n})P(\boldsymbol{n},t) + \\
&+ \mathbb{T}(\boldsymbol{n}|n_T^\alpha + 1, n_D^\alpha - 1)P(n_T^\alpha + 1, n_D^\alpha - 1;t) - \mathbb{T}(n_T^\alpha + 1, n_D^\alpha - 1|\boldsymbol{n})P(\boldsymbol{n},t) + \\
&+ \mathbb{T}(\boldsymbol{n}|n_D^\alpha - 1, n_S^\alpha + 1)P(n_D - 1^\alpha, n_S^\alpha + 1;t) - \mathbb{T}(n_D^\alpha - 1, n_S^\alpha + 1|\boldsymbol{n})P(\boldsymbol{n},t) + \\
&+ \mathbb{T}(\boldsymbol{n}|n_S^\alpha - 1)P(n_S^\alpha - 1;t) - \mathbb{T}(n_S^\alpha - 1|\boldsymbol{n})P(\boldsymbol{n},t) + \\
&+ \mathbb{T}(\boldsymbol{n}|n_D^\alpha + 1, n_S^\alpha - 1)P(n_D^\alpha + 1, n_S^\alpha - 1;t) - \mathbb{T}(n_D^\alpha + 1, n_S^\alpha - 1|\boldsymbol{n})P(\boldsymbol{n},t) + \\
&+ \mathbb{T}(\boldsymbol{n}|n_T^\alpha + 1)P(n_T^\alpha + 1;t) - \mathbb{T}(n_T^\alpha + 1|\boldsymbol{n})P(\boldsymbol{n},t) + \\
&+ \mathbb{T}(\boldsymbol{n}|n_S^\alpha + 1)P(n_S^\alpha + 1;t) - \mathbb{T}(n_S^\alpha + 1|\boldsymbol{n})P(\boldsymbol{n},t) + \\
\\
&+ \sum_{\beta=1}^{\Omega} W_{\alpha\beta}[\mathbb{T}(\boldsymbol{n}|n_T^\alpha + 1, n_T^\beta - 1)P(n_T^\alpha + 1, n_T^\beta - 1;t) - \mathbb{T}(n_T^\alpha + 1, n_T^\beta - 1|\boldsymbol{n})P(\boldsymbol{n},t) + \\
&+ \mathbb{T}(\boldsymbol{n}|n_D^\alpha + 1, n_D^\beta - 1)P(n_D^\alpha + 1, n_D^\beta - 1;t) - \mathbb{T}(n_D^\alpha + 1, n_D^\beta - 1|\boldsymbol{n})P(\boldsymbol{n},t) + \\
&+ \mathbb{T}(\boldsymbol{n}|n_S^\alpha + 1, n_S^\beta - 1)P(n_S^\alpha + 1, n_S^\beta - 1;t) - \mathbb{T}(n_T^\alpha + 1, n_T^\beta - 1|\boldsymbol{n})P(\boldsymbol{n},t) + \\
&+ \mathbb{T}(\boldsymbol{n}|n_T^\alpha - 1, n_T^\beta + 1)P(n_T^\alpha - 1, n_T^\beta + 1;t) - \mathbb{T}(n_T^\alpha - 1, n_T^\beta + 1|\boldsymbol{n})P(\boldsymbol{n},t) + \\
&+ \mathbb{T}(\boldsymbol{n}|n_D^\alpha - 1, n_D^\beta + 1)P(n_D^\alpha - 1, n_D^\beta + 1;t) - \mathbb{T}(n_D^\alpha - 1, n_D^\beta + 1|\boldsymbol{n})P(\boldsymbol{n},t) + \\
&+ \mathbb{T}(\boldsymbol{n}|n_S^\alpha - 1, n_S^\beta + 1)P(n_S^\alpha - 1, n_S^\beta + 1;t) - \mathbb{T}(n_T^\alpha - 1, n_T^\beta + 1|\boldsymbol{n})P(\boldsymbol{n},t)]\}
\end{aligned}
\tag{14}
$$

The master equation can be written in a more compact form though the use of the step operators $\epsilon_i^{\pm(\alpha)}$ that act on a generic function $g(\boldsymbol{n})$ as $\epsilon_i^{\pm(\alpha)}g(n_i^\alpha) = g(n_i^\alpha \pm 1) \quad \forall i \in \{\mathrm{T}, \mathrm{D}, \mathrm{S}\}$ and $\alpha = 1, \ldots, \Omega$:

$$
\begin{aligned}
\frac{\partial P(\boldsymbol{n},t)}{\partial t} =& \sum_{\alpha=1}^{\Omega} \{[(\epsilon_T^{-(\alpha)} - 1)\mathbb{T}(n_T^\alpha + 1|\boldsymbol{n}) + (\epsilon_S^{-(\alpha)} - 1)\mathbb{T}(n_S^\alpha + 1|\boldsymbol{n}) + \\
&+ (\epsilon_T^{+(\alpha)} - 1)\mathbb{T}(n_T^\alpha - 1|\boldsymbol{n}) + (\epsilon_T^{+(\alpha)}\epsilon_D^{-(\alpha)} - 1)\mathbb{T}(n_T^\alpha - 1, n_D^\alpha + 1|\boldsymbol{n}) + \\
&+ (\epsilon_T^{-(\alpha)}\epsilon_D^{+(\alpha)} - 1)\mathbb{T}(n_T^\alpha + 1, n_D^\alpha - 1|\boldsymbol{n}) + (\epsilon_D^{+(\alpha)}\epsilon_S^{-(\alpha)} - 1)\mathbb{T}(n_D^\alpha - 1, n_S^\alpha + 1|\boldsymbol{n}) + \\
&+ (\epsilon_S^{+(\alpha)} - 1)\mathbb{T}(n_S^\alpha - 1|\boldsymbol{n}) + (\epsilon_D^{-(\alpha)}\epsilon_S^{+(\alpha)} - 1)\mathbb{T}(n_D^\alpha + 1, n_S^\alpha - 1|\boldsymbol{n})]P(\boldsymbol{n},t) + \\
&+ \sum_{\beta=1}^{\Omega} W_{\alpha\beta}[(\epsilon_T^{+(\alpha)}\epsilon_T^{-(\beta)} - 1)\mathbb{T}(n_T^\alpha - 1, n_T^\beta + 1|\boldsymbol{n}) + \\
&+ (\epsilon_D^{+(\alpha)}\epsilon_D^{-(\beta)} - 1)\mathbb{T}(n_D^\alpha - 1, n_D^\beta + 1|\boldsymbol{n}) + \\
&+ (\epsilon_S^{+(\alpha)}\epsilon_S^{-(\beta)} - 1)\mathbb{T}(n_S^\alpha - 1, n_S^\beta + 1|\boldsymbol{n})]P(\boldsymbol{n},t)\}
\end{aligned}
\tag{15}
$$

The master equation cannot be solved analytically, and therefore, we apply the van Kampen system-size expansion (*van Kampen, 2014*) to analyze both the deterministic behavior and the role of demographic noise. For this purpose, we split each discrete variable $n_X^\alpha(t)$ into two contributions, the deterministic one, which is obtained taking the limit $N \to \infty$ denoted by $\phi_X^{(\alpha)}$, and the fluctuations $\xi_X^{(\alpha)}$ whose amplitude scales as $1/\sqrt{N}$, as

$$\frac{n_T^\alpha(t)}{N} = \phi_T^\alpha(t) + \frac{\xi_T^\alpha}{\sqrt{N}}$$

$$\frac{n_D^\alpha(t)}{N} = \phi_D^\alpha(t) + \frac{\xi_D^\alpha}{\sqrt{N}} \qquad \alpha = 1,\dots,\Omega \qquad (16)$$

$$\frac{n_S^\alpha(t)}{N} = \phi_S^\alpha(t) + \frac{\xi_S^\alpha}{\sqrt{N}}$$

We introduce a new probability function $\Pi(\boldsymbol{\xi},t) \equiv P(\boldsymbol{n}(\phi(t),\boldsymbol{\xi});t)$, where the $\boldsymbol{\xi}$ is a $3\Omega$-dimensional vector whose components are

$$\boldsymbol{\xi} = (\xi_T^1,\xi_D^1,\xi_S^1,\dots,\xi_T^\Omega,\xi_D^\Omega,\xi_S^\Omega) \equiv (\boldsymbol{\xi}^1,\dots,\boldsymbol{\xi}^\Omega) \qquad (17)$$

so that the left-hand side of the master *Equation (15)* can be recast in the form

$$\frac{\partial P}{\partial t} = \sum_{\alpha=1}^{\Omega} \left[ \frac{1}{N}\frac{\partial \Pi}{\partial \tau} - \frac{1}{\sqrt{N}}\sum_{i\in\{\mathrm{T,D,S}\}} \frac{\partial \Pi}{\partial \xi_i^\alpha}\frac{d\phi_i^\alpha}{d\tau} \right] \qquad (18)$$

with $\tau = t/N$.

Operating an expansion of the step operators with respect to the small parameter $1/\sqrt{N}$, performing the change of variable (*Equation 16*) into the right-hand side of the master equation, and collecting terms proportional to $1/\sqrt{N}$, we get

$$\frac{d\phi_i^\alpha}{d\tau} = h_i^{(\alpha)}(\phi^\alpha) + \nu \sum_{\beta=1}^{\Omega} \Delta_{\alpha\beta}\phi_i^\beta \qquad (19)$$

where

$$h_T^{(\alpha)}(\phi^\alpha) = \phi_U^\alpha k_{UT} + \phi_D^\alpha k_{DT} - \phi_T^\alpha k_{TU} - \phi_T^\alpha k_{TD} \qquad (20)$$

$$h_D^{(\alpha)}(\phi^\alpha) = \phi_T^\alpha k_{TD} + \phi_S^\alpha k_{SD} - \phi_D^\alpha k_{DT} - \phi_D^\alpha k_{DS} \qquad (21)$$

$$h_S^{(\alpha)}(\phi^\alpha) = \phi_U^\alpha k_{US} + \phi_D^\alpha k_{DS} - \phi_S^\alpha k_{SU} - \phi_S^\alpha k_{SD} \qquad (22)$$

and the matrix $\Delta$ denotes the discrete Laplacian operator. The concentration $\phi_U$ reads $[\mathrm{KaiC}] - \phi_T - \phi_D - \phi_S$ and $[\mathrm{KaiC}]\phi_X \to \phi_X$ for the ease of notation.

## Stability analysis

We set to zero the coupling parameter $\nu$, which amounts to neglecting the spatial terms. In this way, system (*Equation 19*) simplifies to

$$\frac{d\phi_i}{d\tau} = h_i(\phi^\alpha) \qquad (23)$$

We fix all the parameters to the values specified in *Appendix 1—table 1*, while [KaiA] and $\gamma$ are used as control parameters and denote by $\phi^* = (\phi_T^*,\phi_D^*,\phi_S^*)$ the homogeneous stationary solution of system (*Equation 23*). To characterize the nature of the fixed points, we perform a linear stability analysis. For this purpose, for each positive $\phi^*$ we evaluate the Jacobian matrix $J$

$$J = \begin{pmatrix} -k_{UT} - k_{TD} - k_{TU} & -k_{UT} + k_{DT} & J1 \\ k_{TD} & -k_{DT} - k_{DS} & J2 \\ -k_{US} & -k_{US} + k_{DS} & J3 \end{pmatrix} \qquad (24)$$

where

$$
\begin{aligned}
J1 &= -k_{UT} + f'(\phi_S^*)[\phi_U^* k_{UT}^A + \phi_T^*(-k_{TU}^A - k_{TD}^A) + \phi_D^* k_{DT}^A] \\
J2 &= k_{SD} + f'(\phi_S^*)[\phi_T^* k_{TD}^A + \phi_D^*(-k_{DT}^A - k_{DS}^A) + \phi_S^* k_{SD}^A] \\
J3 &= -k_{US} - k_{SD} - k_{SU} + f'(\phi_S^*)[\phi_U^* k_{US}^A + \phi_D^* k_{DS}^A + \phi_S^*(-k_{SU}^A - k_{SD}^A)]
\end{aligned}
$$

and calculate its three eigenvalues. The nature of the equilibrium points and their stability is summarized in *Appendix 1—figure 2*, where we plot the equilibrium points to varying $\gamma$. Continuous lines denote stable equilibrium points, while dashed lines denote unstable solutions. The diagram shows three different regions: in region I, the system admits only one stable complex equilibrium point that becomes unstable as long as $\gamma > \gamma_1$. The value $\gamma_1$ marks a supercritical Hopf bifurcation though which we enter in region II. Here, we have a couple of complex eigenvalues with positive real part and the trajectories approach a stable limit cycle. For $\gamma > \gamma_2$, the system undergoes a saddle-node bifurcation with the sudden creation of two additional fixed points, one stable and the other unstable (region III).

**Appendix 1—table 2.** Oligodeoxynucleotide primers used in this work.

| Name | Sequence 5′–3′ |
|---|---|
| alr0523-EcoRI-Fw | TTTTGAATTCGCTTATAAACAGCAGTTAACAGGCT |
| alr0523-Rev | TGCTACCTCCACCGCCTGCCTGTTCAACTACTTTGGA |
| 4G-GFP-Fw | GCGGTGGAGGTAGCAAAGGAGAAGAACTTTTCAC |
| GFP-Rev | GCCTGAATTCTTATTTGTATAGTTCATCCATGCC |
| alr0523-1-Fw | GAATTCGCTTATAAACAGCAGTTAACAGGCT |
| alr0523-1-Rev | CTAGCACCTCCACCGCCTGCCTGTTCAACTACTTTGGA |
| 4G-GFP-Fw in plasmid PCSV3 | ATTTGAAACTGCGCCACGGATC |
| Rev plasmid PCSV3 | GACCATGACGGATTAGCTCAGTAG |
| alr0523(7120)–1 | CGT GAG TCT CCA ACG GAG GC |
| Kai-1 | GAAACTGCAGGCAGAATAGGAAATCTCTAC |
| Kai-2 | CCAAATGATATCGTGCTGACAAACCTACAGTGC |
| Kai-3 | CAGCACGATATCATTTGGTATCGTACTATATTC |
| Kai-4 | CTTTCTGCAGGTTGTCCAGCCAGCAGGGTAG |
| kaiA-2 | CAGGGTGAGGCGATAATCCAT |
| kaiA-1 | GCCAGAGTACTTGTTTCTAAGCAAC |
| CK1-R | CGATTCCGAAGCCCAACCT |
| kaiC-4 | CGAGCTACCAACCGAAAG |
| kaiB-1 | CGGCAATACTCCAAACTCAG |
| PkaiBC-1 | GGTCTATCCCACGAGAAACC |
| YFP-2 | GGTAGCTCAGGTAGTGGTTGTC |
| all5167-1q (forward) | GCTCAAGCAATTCGTCACTGTTCC |
| all5167-2q (reverse) | AAAGATTGCGTCGGTCTGGTGT |
| rnpB-1q (forward) | CTCTTGGTAAGGGTGCAAAGGTG |
| rnpB-4q (reverse) | GGCTCTCTGATAGCGGAACTGG |
| kaiC-3q (forward) | ATGAAGCAGTGGGAGTGGTG |
| kaiC-8q (reverse) | ACGTTACGGGCTATGACCAC |
| kaiB-1q (forward) | ACCAAATTCAGTCAGGGCGT |
| kaiB-4q (reverse) | GCCAATCAGAACTCTTTCCCG |
| kaiA-3q (forward) | CAACTCAAATCAGATTATCGCCA |
| kaiA-4q (reverse) | CTGCCCTCTAGTCGTAGCTG |
| pecB-1q (forward) | ATATTTAATGCTGGTGGTGCTTGTT |

*Continued on next page*

*Appendix 1—table 2 continued*

| Name | Sequence 5′–3′ |
| --- | --- |
| pecB-4q (reverse) | GCAGCGATCGTCCATGACACTAC |
| rpaA-1q (forward) | TTTAACGCCGGAGCAGATGA |
| rpaA-4q (reverse) | TGTCCGTGACGTTGTAGCAA |

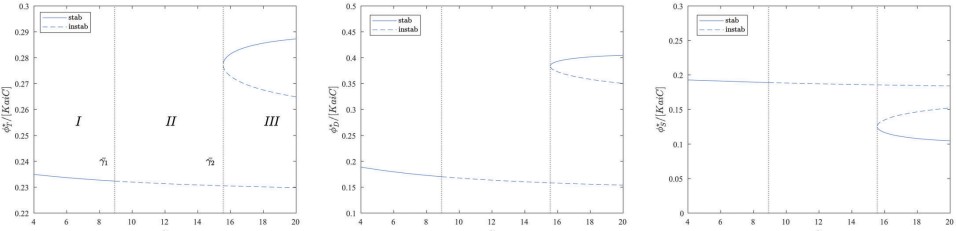

**Appendix 1—figure 2.** Stability diagram. Stability of the equilibrium points for the three species as a function of $\gamma$ and $[KaiA]$. The values of all the other parameters are specified in *Appendix 1—table 1*. Continuous lines denote stable equilibrium points, and dashed lines denote unstable equilibrium points.

To better investigate the region of the parameter's space where the deterministic system exhibits regular oscillations similar to those observed in experiments, we focused on an interval of $\gamma$ including the two critical points $\gamma_1$ and $\gamma_2$. In *Appendix 1—figure 3*, we delimit with blue line the portion of the parameter plane ($\gamma$,$[KaiA]$) where the concentrations of three phosphoforms of KaiC exhibit regular oscillations. The period of oscillations corresponding to the smaller selected region delimited by the black rectangle is reported in *Figure 3B*. From *Figure 3B*, it appears that the region with period $T \simeq 24$ hr is close to the separatrix between the stable fixed point dynamics and the limit cycle regions. A typical result of a numerical integration of system (*Equation 23*) for a set of parameters outside the limit cycle region, but close to the border, is shown in *Appendix 1—figure 4*. Concentrations exhibit damped oscillations that eventually reach the stationary state. Performing stochastic simulations for the same parameters as those in *Appendix 1—figure 4* yields the results displayed in *Appendix 1—figure 5*. Quasi-regular oscillations develop when the deterministic solutions attain the corresponding equilibrium value (*McKane and Newman, 2005*). To shed light onto this phenomenon is entirely devoted the forthcoming discussion.

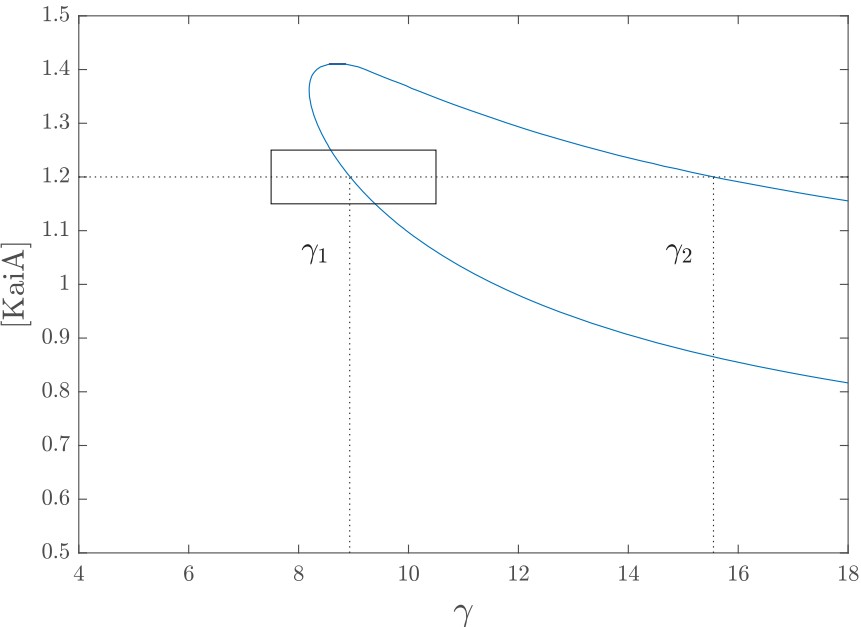

**Appendix 1—figure 3.** Deterministic limit cycle region. The portion of the plane delimited by the blue line marks the region of the parameter's space where deterministic regular oscillations occur.

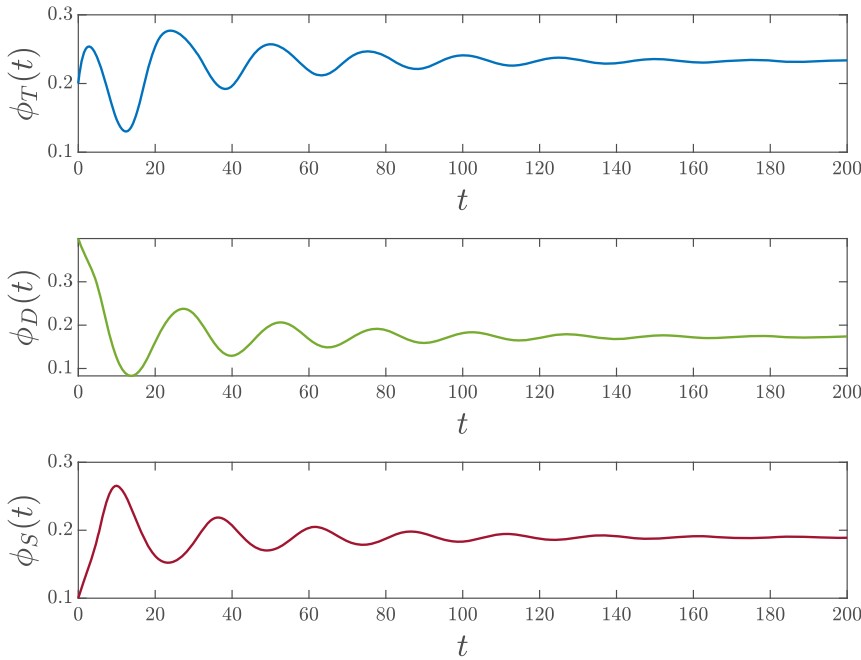

**Appendix 1—figure 4.** Deterministic simulation. Results of the numerical integration of system (*Equation 23*) for $\gamma = 8$ and $[KaiA] = 1.2$. The other parameters are assigned as specified in *Appendix 1—table 1*. $\phi_X$ for $X = T, D, S$ stands here for the relative abundance of the phosphoforms in units of $[KaiC]$.

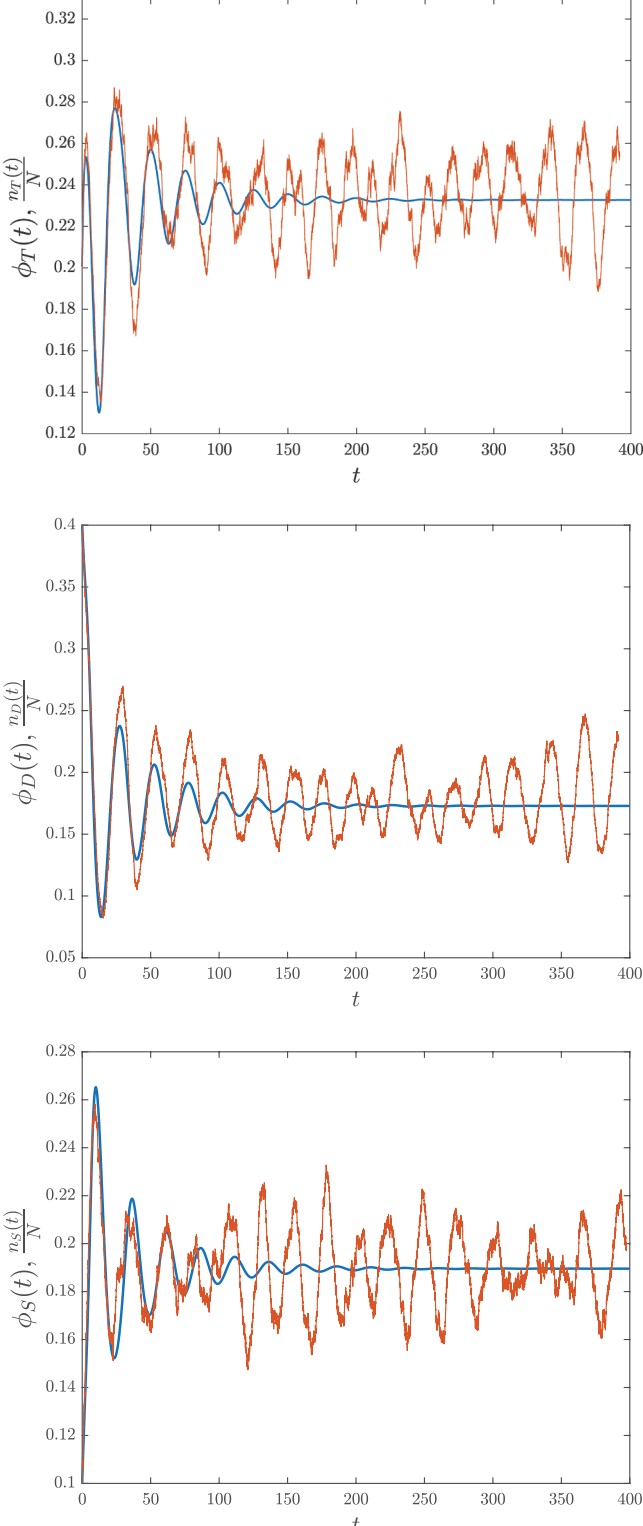

**Appendix 1—figure 5.** Comparison between deterministic and stochastic simulations. The blue lines denote the result of the numerical integration of *Equation (23)* while the noisy red lines represent the stochastic simulation of the system through the *Gillespie, 1977* algorithm. For all the plots for

*Appendix 1—figure 5 continued on next page*

*Appendix 1—figure 5 continued*

$\gamma = 8$ and $[KaiA] = 1.2$ while the other parameters are set as those in *Appendix 1—table 1*. $\phi_X$ for $X = T, D, S$ stands here for the relative abundance of the phosphoforms in units of [KaiC].

## Analysis of fluctuations

Considering the next-to-leading order of the van Kampen expansion and collecting together terms proportional to $1/N$, we find the differential equation for the probability distribution $\Pi(\boldsymbol{\xi}, \tau)$, which results in the following Fokker–Planck equation:

$$\frac{\partial \Pi(\boldsymbol{\xi}, t)}{\partial \tau} = \sum_{\alpha=1}^{\Omega} \left\{ -\sum_{i=1}^{3} \frac{\partial}{\partial \xi_i^\alpha} [\mathcal{A}_i^\alpha(\boldsymbol{\xi}) \Pi(\boldsymbol{\xi}, \tau)] + \frac{1}{2} \sum_{i,j=1}^{3} \sum_{\beta=1}^{\Omega} \frac{\partial^2}{\partial \xi_i^\alpha \partial \xi_j^\beta} [\mathcal{B}_{ij,\alpha\beta} \Pi(\boldsymbol{\xi}, \tau)] \right\} \tag{25}$$

where $\mathcal{A}$ is

$$\mathcal{A}_i^\alpha = \sum_{j=1}^{3} \sum_{\beta=1}^{\Omega} \mathcal{M}_{ij,\alpha\beta} \xi_j^\beta \tag{26}$$

The two matrices $\mathcal{M}$ and $\mathcal{B}$ have dimension $3\Omega \times 3\Omega$, and they can be expressed as the sum of two distinct terms, one due to the spatial contribution that we label with the superscript $(SP)$, and the other related to the interactions within each cell that we label as $(NS)$. Moreover, we assume that all the entries of the two matrices are evaluated at the equilibrium point $\phi^*$. Thus, we have

$$\mathcal{M}_{ij,\alpha\beta} = \mathcal{M}_{ij}^{(NS)} \delta_{\alpha\beta} + \mathcal{M}_{ij}^{(SP)} \Delta_{\alpha\beta} \tag{27}$$

with

$$\mathcal{M}^{(NS)} = J \tag{28a}$$

$$\mathcal{M}^{(SP)} = \begin{pmatrix} \nu & 0 & 0 \\ 0 & \nu & 0 \\ 0 & 0 & \nu \end{pmatrix} \tag{28b}$$

where $J$ is the Jacobian matrix relative to system (*Equation 19*) evaluated at the steady state. For $\mathcal{B}$, we have

$$\mathcal{B}_{ij,\alpha\beta} = \mathcal{B}_{ij}^{(NS)} \delta_{\alpha\beta} + \mathcal{B}_{ij}^{(SP)} \Delta_{\alpha\beta} \tag{29}$$

with

$$\mathcal{B}^{(NS)} \equiv \begin{pmatrix} \phi_U^* k_{UT} + \phi_D^* k_{DT} + & -\phi_T^* k_{TD} - \phi_D^* k_{DT} & 0 \\ +\phi_T^* k_{TU} + \phi_T^* k_{TD} & & \\ & & \\ -\phi_T^* k_{TD} - \phi_D^* k_{DT} & \phi_T^* k_{TD} + \phi_S^* k_{SD} + & -\phi_D^* k_{DS} - \phi_S^* k_{SD} \\ & +\phi_D^* k_{DT} + \phi_D^* k_{DS} & \\ & & \\ 0 & -\phi_D^* k_{DS} - \phi_S^* k_{SD} & \phi_U^* k_{US} + \phi_D^* k_{DS} + \\ & & +\phi_S^* k_{SU} + \phi_S^* k_{SD} \end{pmatrix} \tag{30a}$$

$$\mathcal{B}^{(SP)} = \begin{pmatrix} -2\nu\phi_T^* & 0 & 0 \\ 0 & -2\nu\phi_D^* & 0 \\ 0 & 0 & -2\nu\phi_S^* \end{pmatrix} \tag{30b}$$

## Spatial correlations

The Fokker–Planck *Equation (25)* describes the fluctuations about the equilibrium point $\phi^*$, and it is equivalent to the following coupled Langevin equations:

$$\frac{d\xi_i^\alpha}{d\tau} = \sum_{j=1}^{3}\sum_{\beta=1}^{\Omega} \mathcal{M}_{ij,\alpha\beta}\xi_j^\beta + \eta_i^\alpha \tag{31}$$

where $\xi_i^\alpha$ is the component of vector $\boldsymbol{\xi}^\alpha$ defined in *Equation (17)* while $\eta_i^\alpha$ is the $i$th component of the noise $\eta^\alpha$ with the following properties:

$$\langle \eta_i^\alpha(\tau) \rangle = 0 \tag{32}$$

$$\langle \eta_i^\alpha(\tau)\eta_j^\beta(\tau') \rangle = \mathcal{B}_{ij,\alpha\beta}\delta(\tau - \tau') \tag{33}$$

Starting from this equation, it is possible to derive an analytic expression for the so-called power spectrum density matrix (PSDM), a diagnostic tool that allows to highlight the presence of correlations between species at different distances along the chain (*Challenger and McKane, 2013*; *Zankoc et al., 2019*; *Zankoc et al., 2017*). The (PSDM) $P_{ij,\alpha\beta}(\omega)$ is defined by

$$P_{ij,\alpha\beta}(\omega) = \langle \tilde{\xi}_i^\alpha(\omega)\tilde{\xi}_j^{\beta*}(\omega) \rangle \tag{34}$$

where $\tilde{\xi}_i^\alpha(\omega)$ is the temporal Fourier transform of the $i$th component of $\boldsymbol{\xi}^\alpha(\tau)$. The degree of correlation between species $i$ in cell $\alpha$ and species $j$ in cell $\beta$ can therefore be quantified through the complex coherence function (CCF), a $3\Omega \times 3\Omega$ matrix $C$, whose entries are the ratio of the elements of PSDM to a normalization coefficient:

$$C_{ij,\alpha\beta}(\omega) = \frac{P_{ij,\alpha\beta}}{\sqrt{P_{ii,\alpha\alpha}P_{jj,\beta\beta}}} \tag{35}$$

To obtain an analytic expression of $P_{ij,\alpha\beta}$, we solve the Langevin *Equation (31)* performing a temporal Fourier transform. In this way, for each component of the fluctuations vector on cell $\alpha$ we get

$$\tilde{\xi}_i^\alpha(\omega) = \sum_{j=1}^{3}\sum_{\beta=1}^{\Omega} (-i\omega\delta_{ij,\alpha\beta} - \mathcal{M}_{ij,\alpha\beta})^{-1}\eta_j^\beta(\omega) = \sum_{j=1}^{3}\sum_{\beta=1}^{\Omega} \Phi_{ij,\alpha\beta}^{-1}(\omega)\eta_j^\beta(\omega) \tag{36}$$

with $\Phi(\omega) = -i\omega\mathbb{1} - \mathcal{M}$, where $\mathbb{1}$ denotes the $3\Omega \times 3\Omega$ identity matrix. Substituting *Equation (36)* into *Equation (34)*, one eventually obtains

$$\begin{aligned}
P_{ij,\alpha\beta}(\omega) &= \langle \tilde{\xi}_i^\alpha(\omega)\tilde{\xi}_j^{\beta*}(\omega) \rangle = \\
&= \sum_{l,m=1}^{3}\sum_{\gamma,\delta=1}^{\Omega} \Phi_{il,\alpha\gamma}^{-1}(\omega)\langle \tilde{\eta}_l^\gamma(\omega)\tilde{\eta}_m^{*\delta}(\omega) \rangle (\Phi_{mj,\delta\beta}^\dagger)^{-1}(\omega) = \\
&= \sum_{l,m=1}^{3}\sum_{\gamma,\delta=1}^{\Omega} \Phi_{il,\alpha\gamma}^{-1}(\omega)(\mathcal{B}_{lm,\gamma\delta})(\Phi_{mj,\delta\beta}^\dagger)^{-1}(\omega) = \\
&= (\Phi^{-1}\mathcal{B}(\Phi^\dagger)^{-1})_{ij,\alpha\beta}
\end{aligned} \tag{37}$$

where $\Phi^\dagger = (\Phi^*)^T$ is the complex conjugate transpose of $\Phi$ and use has been made of the following properties:

$$\langle \tilde{\eta}_i^\alpha(\omega) \rangle = 0 \tag{38a}$$

$$\langle \tilde{\eta}_i^\alpha(\omega)\tilde{\eta}_j^\beta(\omega') \rangle = \mathcal{B}_{ij,\alpha\beta} \tag{38b}$$

## Accounting for a smooth kernel of interactions

In this section, we will extend the analysis so as to account for a generalized kernel of interaction, which decays with the inter-particle distance. More concretely, each node is coupled with all the other nodes along the chain, but the strength of the coupling is modulated as a monotonous decreasing function of the distance, namely, $f(|\alpha - \beta|)$. Adjacent nodes are hence prone to mutual influences as compared to nodes that are far apart from each other. The newly imposed modulation reflects in the transition rates (12) as follows:

$$T(n_i^\alpha - 1, n_i^\beta + 1|\boldsymbol{n}) = \frac{n_i}{N}\nu f(|\alpha - \beta|) \tag{39}$$

where $\nu$ is a constant and $f(|\alpha - \beta|)$ is a decreasing function of the distance between nodes $\alpha$ and $\beta$. This results in a modification of the diffusive-like term in the master equation, which can be reabsorbed in a new definition of the adjacency matrix $W_{\alpha\beta}$ as

$$W'_{\alpha\beta} = W_{\alpha\beta}f(|\alpha - \beta|) \tag{40}$$

and the corresponding Laplacian matrix $\Delta'$ now becomes

$$\Delta'_{\alpha\beta} = W'_{\alpha\beta} - k_\alpha\delta_{\alpha\beta} \tag{41}$$

Accordingly, the two matrices (*Equations 27* and *29*) appearing in the Fokker–Planck *Equation (25)* must be replaced by

$$\mathcal{M}_{ij,\alpha\beta} = \mathcal{M}_{ij}^{(NS)}\delta_{\alpha\beta} + \mathcal{M}_{ij}^{(SP)}\Delta'_{\alpha\beta} \tag{42}$$

$$\mathcal{B}_{ij,\alpha\beta} = \mathcal{B}_{ij}^{(NS)}\delta_{\alpha\beta} + \mathcal{B}_{ij}^{(SP)}\Delta'_{\alpha\beta} \tag{43}$$

In the following, we will focus on two different choices for the function $f$. Our first choice is to consider an exponentially decaying kernel

$$f(|\alpha - \beta|) = e^{-\sigma|\alpha-\beta|} \tag{44}$$

were $\sigma > 0$. The second option that we set to explore assumes a power-law decay in the form

$$f(|\alpha - \beta|) = \frac{1}{|\alpha - \beta|^\rho} \tag{45}$$

with $\rho > 0$. In the next section, we will challenge different scenarios against experimental data.

## Comparison with experiments

Setting $\nu = 0$ in *Equation (37)* and considering the diagonal terms $\alpha = \beta$ and $i = j$ returns $N$ identical temporal power spectra of the fluctuations. The analytical prediction can be used to fit experimental data for both *Anabaena* (see *Figure 4C*) and *Synechococcus* (see *Figure 3C*). In both cases, the values of $\gamma$ and [KaiA] predicted by the fit lay outside the region where deterministic oscillations occur.

Further, the CCF can be invoked to quantitatively explain the correlation of the fluorescence signal recorded in different cells of the *Anabaena* filaments. More specifically, we will adjust the theoretically predicted CCF to the analogous quantity estimated experimentally by varying the coupling parameter $\nu$ and the connectivity $k_\alpha$ when postulating a uniform range of interaction or by varying $\nu$ and the characteristic parameter of the decay function $f(|\alpha - \beta|)$, a smoothly decaying kernel of interaction.

To calculate the experimental spatial correlation, we proceeded as follows. We focused on a given portion made of $\Omega$ cells of the full *Anabaena* filament. We recall that this latter is growing in time. Hence, as time progresses, the cells within the examined portion of filament get replaced by their offspring as follows cellular duplication. However, for the analysis of spatial correlations, we are only interested in the relative positioning of the cells. In other words, we analyze the fluorescence content of each cell in relation to the position occupied within the filament irrespectively of the fact that the monitored population of $\Omega$ cells is constantly renewed in time by new offspring generation.

By following this procedure, we obtained Ω time series for the fluorescence on each spot along the chain. Then these temporal signals are transformed via a standard FFT in time. We further select the signal corresponding to $\omega^*$, the frequency at which the temporal power spectrum exhibits a peak, and insert it in *Equation (34)*.

The results of the comparison between theory and experiments are plotted in *Figure 5* and *Appendix 1—figure 6*. The plot in the main text corresponds to the exponential long-range cell-cell communication (*Equation 44*), where the best-fit parameters are $\nu = 0.61$ and $\sigma = 0.35$. Notice that $1/\sigma \simeq 3$ set the characteristic scale of the spatial interaction (in units of cells). In *Appendix 1—figure 6*, we compare the fit of experimental data (red circles) obtained by using a fixed interaction kernel with connectivity $k_\alpha = 8$ (black squares) and a decaying function (*Equation 45*) (blue squares). In all cases, the experimental points are nicely interpolated by the theoretical curves. Accounting for a smooth kernel of interaction yields more accurate fittings.

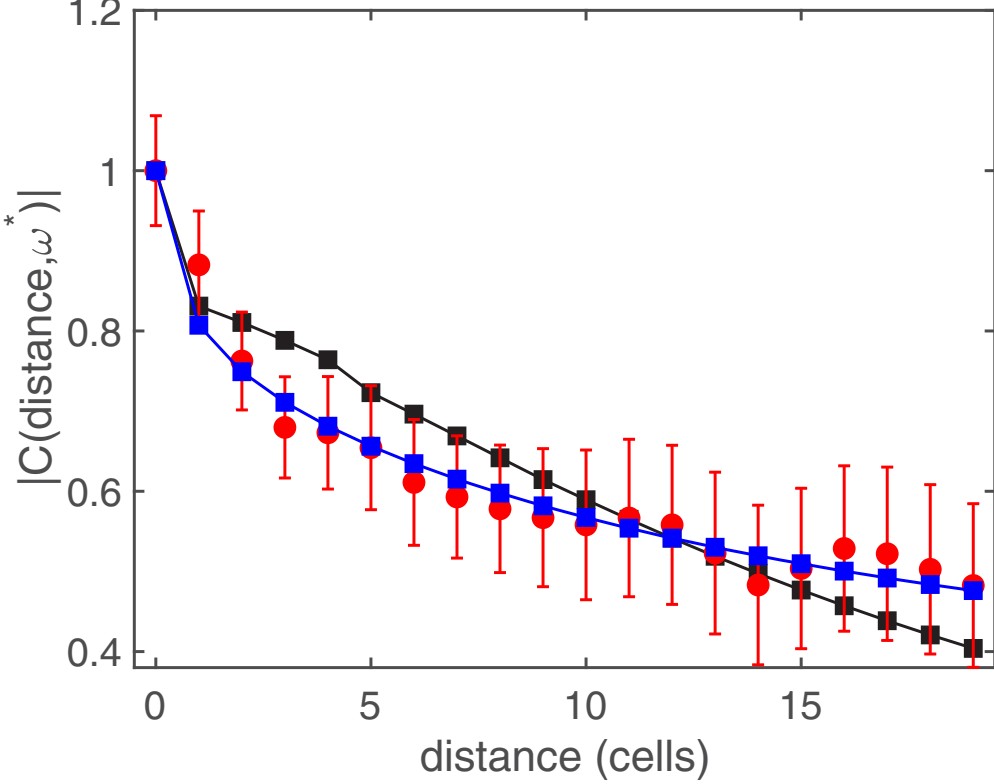

**Appendix 1—figure 6.** The complex coherence function: comparison between theory and experiments. Complex coherence function measuring the correlation of 35 cell segments at the frequency of temporal oscillations. Red circles correspond to experimental data, and the squares represent fits to the experimental data with the prediction of the stochastic model using the constant kernel (black squares) and the power-law kernel (blue squares). Lines between symbols are a guide to the eye. The fitting procedure gives $\nu = 0.49$ for the fixed connectivity $k_\alpha = 8$, while it returns $\nu = 1.11$ and $\rho = 1.96$ for the power-law function.

## Supplemental methods
### Cloning and strain construction

For construction of a fusion of the promoter of *pecB* to the gfp-mut2 gene in *Anabaena* sp. (also known as Nostoc sp.) strain PCC 7120, a custom DNA sequence containing an EcorRI restriction site preceded by the pecB sequence of *Anabaena* was synthesized and inserted into the pUC57 plasmid (Bio Basic) (region alr0523). The *pecB* construct was amplified with the primer pair alr0523-EcoRI-Fw/alr0523-Rev. A pCSAL33 plasmid, which bears the *gfp-mut2* sequence, was amplified with the primer pair 4G-GFP-Fw/alr0523-GFP-Rev. The PCR-amplified sequence products gfp-mut2 and pecB

were mixed and amplified with the primer pair (alr0523 PCR-Fw/GFP-PCR2-Rev). The merged PCR product and a CSV3 plasmid were digested with EcoRI and ligated to produce pSVRG4. The transferred sequence was confirmed by sequencing with the primers (alr0523-1-Fw, alr0523-1-Rev, 4G-gfp-Fw in plasmid CSV3, and Rev plasmid CSV3). The final construct bears a sequence of 1337 bp that comprises the *pecB* promoter and the 5′ end of the *pecB* gene linked to the *gfp-mut2* gene by a sequence encoding four glycines. The pSVRG4 plasmid was transferred to *Anabaena* by conjugation, which was performed as described (*Elhai et al., 1997*). Clones resistant to streptomycin (Sm) and spectinomycin dihydrochloride pentahydrate (Sp) were selected, and their genomic structure was tested by PCR (alr0523(7120)–1/GFP-Rev). Periodically performed PCR analysis indicated that the strains were stable, not showing WT chromosomes, in BG11 medium supplemented with antibiotics. The *Anabaena* strains used as recipients in the conjugations were the WT PCC 7120, Δ*sepJ*/Δ*fraCD*, in which *sepJ*, *fraC*, and *fraD* were deleted (*Nürnberg et al., 2015*), and Δ*kaiABC*, in which the *kaiABC* genes were deleted (see *Appendix 1—figure 7*). Strain CSL64 bearing a chromosomally encoded $P_{hetR-gfp}$ transcriptional fusion in a WT background has been reported previously (*Corrales-Guerrero et al., 2013*).

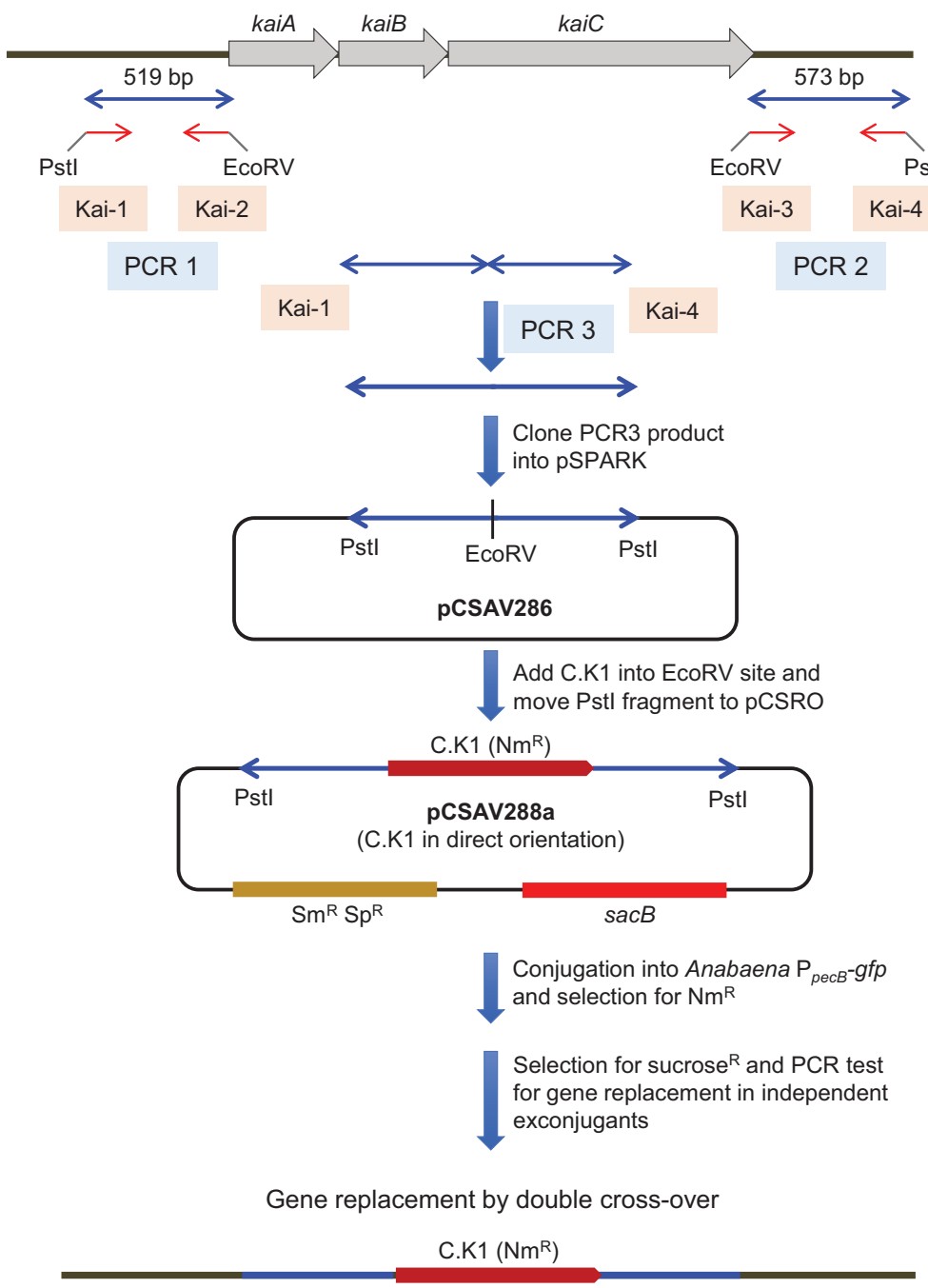

**Appendix 1—figure 7.** Construction of an *AnabaenaΔkaiABC* deletion mutant in the PpecB-gfp genetic background. PCR, DNA restriction/ligation, and transformation into *Escherichia coli* were performed by standard techniques. Conjugation from *E. coli* to *Anabaena* was performed as described by *Elhai et al., 1997*, and sucrose (sacB)-based positive selection for double recombinants was performed as described by *Cai and Wolk, 1990*. *AnabaenaΔkaiABC* homozygous mutants containing the kai deletion with insertion of the C.K1 gene cassette in direct orientation were obtained and confirmed by PCR analysis. Kai-1 to Kai-4 are oligodeoxynucleotide primers.

The *Synechococcus* strain containing the gene encoding a YFP-SsrA reporter whose expression is driven by the *kaiBC* promoter at neutral site II was constructed by transforming *Synechococcus* sp.

PCC 7942 with plasmid EB2316, which was a gift from Erin O'Shea (Addgene plasmid # 87753; http://n2t.net/addgene:87753; RRID:Addgene 87753). The transformation of *Synechococcus* was carried out as previously described (*Golden and Sherman, 1984*). The cells were incubated for 48 hr at 30°C under illumination on nitrocellulose filters, and transformants were selected on Cm-containing BG11 plates. Confirmation of genomic structure was carried out by PCR with the primers kaiC-4, kaiB-1, PkaiBC-1, and YFP-2. All oligodeoxynucleotide sequences used for construction are listed in *Appendix 1—table 2*.

## Culture conditions

Strains and derived strains were grown photoautotrophically in BG11 medium containing $NaNO_3$, supplemented with 20 mM HEPES (pH 7.5) with shaking at 180 rpm, at 30°C, as described previously (*Corrales-Guerrero et al., 2013*; *Di Patti et al., 2018*). Growth took place under constant illumination (10 µmol of photons [spectrum centered at 450 nm]) from a cool-white LED array. When required, Sm and Sp were added to the media at final concentrations of 2 µg/mL for liquid and 5 µg/mL for solid media (1% Difco agar). The densities of the cultures were adjusted so as to have a chlorophyll content of 2–4 µg/mL 24 hr prior to the experiment, following published procedures (*Di Patti et al., 2018*). For time-lapse measurements, filaments in cultures were harvested and concentrated 50-fold. *Synechococcus* cultures were grown as above, and when required, 7.5 µg Cm was added.

## Sequence

### Region alr0523

ttttGAAttcGCTTATAAACAGCAGTTAACAGGCTATTAATAATTATGATGAATATTAGTCTTGAAAA
TTATTATCATTTTTAGCCACAAAATAAAAACCGAAGATTGTTATCTAAAACACAGATTTTTTTTGA
TAAAAATCCTGCTTTAGCTTTGTTTATCAATTTCTACTGTCAGCATGAAAGTATATATAAGCTAGA
TTTGCTACCGCCGTAGAAATTATACTCAAGCATTTATTAAGTAAAGTAAACAATATTATTTAATTG
TTAAACGGTTTTTCACCTAAAGGCATTACAACCTTTATAAAACAATAATTTTTAGAGAAGTCTGTA
TGGATTAACACTATCTTACAAAACTTAATAATAAATTCACTCAAGCCATATATAAAATACGAAC
TAAGGTTTTCAAAGCAATTAAAAAACAGAAAACAAATTGCTGTTTTCCTAATAAGAAATCGCACA
TGATTGCAGTCCCCCAAGATAATATATGGGAGTTAATTCTGCTTGAATATTTGTTTTCTAAATAG
TAAGAATAATTGCAATCGACCTTATAAAAAGCTGCAATGACCTTTAGGAGGAAAGAAAGATGC
TCGATGCTTTTTCCAAAGTAGTTGAACAGGCAggcggtggaggtagca

## Search of putative RpaA regulatory sequences of the *kaiABC, rpaA, pecB,* and *ftsZ* promoter regions in *Anabaena*

The positions for all transcription start sites were extracted from *Mitschke et al., 2011*. For in silico analysis of the *Anabaena* promoter regions, we use a position-specific probability matrix that defines the RpaA binding motif in *Synechococcus* (*Markson et al., 2013*) in FIMO tool (*Grant et al., 2011*).

