## [Decision Letter]

**Acceptance summary:**

This paper applies a combination of experimental and theoretical approaches to study the circadian clock of *Anabaena*, a cyanobacterium which exists as multicellular filaments. The system offers a rare natural example of a one dimensional system of coupled oscillators which have been the subject of much theoretical investigation. The authors demonstrate an interesting role of rpaA as a regulator of the clock and cell-cell communication to synchronize oscillations across cells in the same filament. They also show that inherent demographic noise can expand the parameter range over which oscillations with circadian periods occur, possibly enhancing the robustness of the clock.

**Decision letter after peer review:**

Thank you for submitting your article "Robust, coherent and synchronized circadian clock-controlled oscillations along Anabaena filaments" for consideration by *eLife*. Your article has been reviewed by 3 peer reviewers, and the evaluation has been overseen by a Reviewing Editor and Aleksandra Walczak as the Senior Editor. The following individual involved in review of your submission has agreed to reveal their identity: Mogens H Jensen (Reviewer #2).

The reviewers have discussed the reviews with one another and the Reviewing Editor has drafted this decision to help you prepare a revised submission.

Summary:

All three reviewers found the *Anabaena* system interesting and the analysis of a circadian clock that depends on cell-cell communication to be of general interest. However, they had a series of concerns about the claims made in the paper.

Essential revisions:

1. On the experimental front, please pay particular attention to address points 1a-d raised by Reviewer 1.

2. On the theoretical front, please give a clearer explanation of Figures 5 and 6 (see comments by Reviewers 2 and 3) and provide analyses to show that the claims about the model are robust to parameter choices (points 1 and 2 of Reviewer 3).

Please go through all three reviews appended below and try to address all the comments in a revised manuscript.

We would like to draw your attention to changes in our policy on revisions we have made in response to COVID-19 (https://elifesciences.org/articles/57162). Specifically, when editors judge that a submitted work as a whole belongs in *eLife* but that some conclusions require a modest amount of additional new data, as they do with your paper, we are asking that the manuscript be revised to either limit claims to those supported by data in hand, or to explicitly state that the relevant conclusions require additional supporting data.

Reviewer #1:

Arbel-Goren et al., apply a combination of experimental and theoretical approaches to study the circadian clock of *Anabaena* sp. Unlike *Synechococcus* which is unicellular, *Anabaena* exists as multicellular filaments. The authors follow the expression dynamics of gfp expressed from a clock-controlled promoter (PpecB) and show that expression follows an oscillatory pattern. They find that cells within the same filament (up to 10 cells apart) are synchronous, but cells across unrelated filaments are not. Synchrony is inherited upon division. The oscillatory patten is not consistent with the expression pattern of kai genes. Instead, the authors show that expression is in phase with rpaA expression, but dependent on Kai. To strengthen the idea that cell-cell communication is essential for the oscillations, the authors delete septal proteins, which breaks the pattern observed. Theoretical model of coupled clocks fits the experimental observations. The authors conclude that this clock might provide robustness in case of fluctuating or stressful conditions.

I have focused my comments on the experimental aspects of this manuscript.

1. The main conclusion of the present work is the importance of cell-cell communication in establishing the circadian clock of *Anabaena*, which is in contrast to that observed for the unicellular *Synechococcus*. This is an interesting observation. However, I think the authors must provide more compelling evidence to strengthen this conclusion. For example:

a. The effects of filament lengths are unclear to me. In Figure 2, the authors compare the gene expression profiles for cells up to 10 cells apart in the same filament and state that these are comparable. However, according to their model, there seems to be a length-dependent effect. To test the same, the authors can compare profiles of cells in increasing intervals and provide plots as shown in Figure 2A for the same. In general, while the authors use synchronization index to characterize degree of synchronization in various conditions, the representation of expression profiles as in Figure 2A is better to follow. This analysis for cells deleted for sepJ/ fraCD must be included.

b. In addition to the analysis above, the authors can block cell division (using drug treatment such as cephalexin) to assess how division affects the oscillatory patterns observed.

c. The authors make an argument in the Discussion section that levels of Kai proteins may not be limiting to trigger the coupled clock behaviour they observe. In order to support this conclusion, they must perturb expression of these genes (overexpress the kai operon).

d. The authors propose that this cell-cell communication must involve transport of small molecules. Could the authors please elaborate on how these molecules feed directly into regulation of gene expression patterns? Are they produced in the cell or freely diffusing in the growth media? The current Discussion section does not explain this idea with clarity. It would be nice to include the influence of such a molecule in their theoretical framework as well.

2. The physiological relevance of the mechanism described in this study is unclear to me. How do the authors envision such clock coupling provides robustness and a biological advantage under stress conditions? Could such robustness be assessed under various sugar availabilities or under starvation for example?

Reviewer #2:

This is a nice paper on measurements on circadian oscillations in *Anabaena* filaments. There has been a lot of investigations on circadian rhythms in various systems. I am not an experimentalist and I do not work on circadian systems so it is hard for me to judge whether this paper presents fundamental new measurements on the circadian clock and its relations to the cell cycle. However, the paper appears very comprehensive and presents to the best of my knowledge new nice results.

Let me concentrate on the model presented in Figure 5A. This is defined by different interactions between various phosphorylated states of KaiC gene. I should like the authors to explain in more details the different links in this diagram. It probably makes sense that there are transitions between the phosphorylated states but please explain in more details. It is well know that oscillations are generated from negative feed-back loops.

In the diagram there are two negative links, from S state to Kai A and back again. This by itself defines a positive feed-back loop which will give rise to a switch not to oscillations. Therefore the only negative loop I can identify is the one S -> KaiA -> D(ST) -> S. Is that the loop that is responsible for all the oscillations? I might doubt it but please explain.

The authors have 'exported' all equations to the supporting information but I would have liked whether it is possible to include the equations for this basic loop as a diagram in the figure?

From the underlying deterministic model of this diagram the authors obtain the phase diagram in Figure 5C. I presume the line to the colored region is defined through a Hopf bifurcation, am I right? As the authors mention, there is 'only' a circadian time scale at the lower right boundary. But is that not surprising?

Next, the authors presents a model for an array of coupled circadian clocks which is supposed to model *Anabaena* through cell-cell communications via septul proteins. The full structure of the model is nicely presented in the supporting information. Well, it is not a simple model that is developed and I cannot claim I understand all the involved steps. Some of the results are presented in Figure 6. Figure 6B shows results from Gillespie on how the cells are correlated. Bit I am missing a little more explanation. What is the noise level (volume/number of molecules in Gillespie?), max/min of what. And in the text there is an error where Figure 6B says it is the power spectrum. Altogether, I find it a nice paper.

Reviewer #3:

The system is very interesting and offers a rare natural example of a system of coupled oscillators which have been the subject of much theoretical investigation.

I think the results certainly merit consideration in e*Life*. However, I think some of the theoretical claims made are not sufficiently supported in the paper in its current version.

1. A central argument that the system does not display deterministic oscillations seems to rely heavily on Figure 5C. However, it is a little difficult to see how much that conclusion is robust to the very specific model and parameters used. For example, it would seem that a factor of 2 in [Kai C] and a factor of 10% in [KaiA] would completely change that conclusion. Surely, this is possible given that there is evidence that *Anabaena* differs from *Synechococcus* as the authors themselves claim. Perhaps there is no parameter combination (including parameters other than the two the authors chose to vary) which would show deterministic oscillations without making the Figure 5B fit significantly worse but this is a little unclear from the current text.

2. Following on that point, the authors do not seem to sufficiently discuss the applicability of Rust et al. model and parameters to *Anabaena*. I think the paper would be considerably strengthened by a clear discussion on how robust the conclusion is to different parameter variations and the applicability of this choice, as well as the possibility of deterministic vs noise driven oscillations.

3. There are three different Kernels used for the cell-to-cell communication, about which little is known experimentally. Of these, the exponential seems most plausible in the absence of other knowledge. The caption of Figure 6D says the complex coherence function is fit using an exponential kernel but the sharp cutoff in the fit seems to suggest a constant Kernel with a sharp cutoff. The SI suggests a power law Kernel fits well but it is unclear what the justification of such a Kernel would be. As an aside, is it possible to simplify the equations in a mean-field kind of way?

4. It is a little unclear in Table 1 why R is so high even for different filaments particularly since it was originally introduced in Garcia-Ojalvo as an order parameter that sharply distinguishes between synchronization and lack of.

I think the paper is not suitable for *eLife* in its current form but a clearer and stronger discussion of model choice and fitting could remedy that.

---

## [Author Response]

Reviewer #1:[…] I have focused my comments on the experimental aspects of this manuscript.1. The main conclusion of the present work is the importance of cell-cell communication in establishing the circadian clock of *Anabaena*, which is in contrast to that observed for the unicellular *Synechococcus*. This is an interesting observation. However, I think the authors must provide more compelling evidence to strengthen this conclusion. For example:a. The effects of filament lengths are unclear to me. In Figure 2, the authors compare the gene expression profiles for cells up to 10 cells apart in the same filament and state that these are comparable. However, according to their model, there seems to be a length-dependent effect. To test the same, the authors can compare profiles of cells in increasing intervals and provide plots as shown in Figure 2A for the same. In general, while the authors use synchronization index to characterize degree of synchronization in various conditions, the representation of expression profiles as in Figure 2A is better to follow. This analysis for cells deleted for sepJ/ fraCD must be included.

In all our experiments, filaments were long enough so that any possible end effects were negligible. Consequently, the behavior we observed throughout ALL experiments was essentially independent of filament length. Likewise, in the theoretical model, we first introduced demographic noise for a single oscillator and then generalized to an array of oscillators, without addressing any length dependence. Rather than <milestone-start />“<milestone-end />filament length”, we assume the reviewer meant distance between cells along a single filament. To highlight the high synchronization within filaments, we calculated the synchronization index R in two situations: one in which cells are sufficiently separated (by intervals of at least 10 other cells), and one in which cells are adjacent. As demonstrated by the first two lines in Table 1, both values of R are statistically indistinguishable. We believe that this is compelling evidence for very high synchronization. We also wish to stress that the reviewer<milestone-start />’<milestone-end />s suggestion of plotting individual traces for increasing intervals is already implemented in Figure 2C, where the lineage of two cells are plotted for five cycles. As the reviewer will note, after five cycles cell division results in 32 contiguous cells, whose individual traces largely overlap.

As per the reviewer<milestone-start />’<milestone-end />s suggestion, we have added what is now Figure S1, which includes two panels. In A, we plot the mean fluorescence per cell in a ∆*sepJ*∆*fraC*D filament, as well as the respective autofluorescence, analogously to Figure 2A. Note that the traces extend for only two complete cycles instead of five as in Figure 2A. Deletion of septal junction genes ∆*sepJ*∆*fraC*D turns out to result in filament fragmentation, preventing us from following filaments under the microscope from more than 2-3 days (see also Nurnberg et al., mBio, 2015).

In panel B we plot traces of four adjacent individual cells in a ∆*sepJ*∆*fraC*D filament and their respective lineages, as you requested. The noisier nature of these traces as compared to cell traces in the WT (Figure 2C), highlight the smaller extent of synchronization when cell-cell communication is perturbed, consistently with the smaller value of R in Table 1. The value of R in Table 1 represents the statistics of various such experiments (n=4).

b. In addition to the analysis above, the authors can block cell division (using drug treatment such as cephalexin) to assess how division affects the oscillatory patterns observed.

In our study, we presented evidence that the circadian clock gates cell division, allowing division at some phases but not at others. Whether cell division affects the circadian clock in *Anabaena* is unknown. We point out that in organisms for which the interplay between these two oscillators has been examined (including unicellular *Synechococcus* and eukaryotes), the circadian clock is independent of cell division, and the cell cycle has no effect on the circadian clock (Mori T, Cell division cycles and circadian rhythms. Bact Circadian Programs (2009); Mori et al., J Bacteriol. 183:2439-2444 (2001); Mori et al., PNAS 93:10183 (1996)).

While it would be interesting in future studies to test whether blocking cell division has any effect on the circadian clock of *Anabaena*, cephalexin and similar compounds are not the way to go: cephalexin is an inhibitor of cell wall biosynthesis in *Anabaena* and it would kill growing cells rather than stop cell division.

In this connection, it is important to note that heterocyst cells that develop under nitrogen deprivation in *Anabaena* lose the ability to divide, and yet, the heterocyst-enriched fraction in the bulk experiments of Kushige et al. (J. Bact, 2013) exhibited clear circadian oscillations. This supports the notion that cell division does not affect the circadian clock in *Anabaena*, and a sentence was added to the Discussion about this issue (line 345).

c. The authors make an argument in the Discussion section that levels of Kai proteins may not be limiting to trigger the coupled clock behaviour they observe. In order to support this conclusion, they must perturb expression of these genes (overexpress the kai operon).

We stress that we assumed in our Discussion that the copy number of Kai proteins is likely to be small and unknown (lines 386-396), and we hypothesized that cell-cell communication may compensate for these small numbers in synchronizing clocks. Supporting this hypothesis remains a topic for future studies.

Typically overexpression is carried out and is most informative in cases when deletion of function or mutation (inactivation) proves impossible (e.g. in the case of an essential gene). This is not the case of our experiments, as shown by our results with Δ*kaiABC* filaments. We point out in passing that overexpression of kai genes from plasmids would also be problematic, as cell-cell fluctuations in plasmid copy numbers would introduce an unwanted, extraneous source of variability. In addition, the synchrony we observe in wild-type filaments is so high, that we do not expect that over-expression of Kai proteins will enhance it.

d. The authors propose that this cell-cell communication must involve transport of small molecules. Could the authors please elaborate on how these molecules feed directly into regulation of gene expression patterns? Are they produced in the cell or freely diffusing in the growth media? The current Discussion section does not explain this idea with clarity. It would be nice to include the influence of such a molecule in their theoretical framework as well.

In the present work, we deal exclusively with within-filament communication through septal junctions, and not through freely diffusing molecules in the growth medium. We have made clear in the Discussion that communication between cells takes place through septal junctions (line 297). This said, the precise identity of cell-to-cell signals that couple clocks in different cells has not been established, and remains a topic for future research. Nonetheless, this has been discussed in the first paragraph of the Discussion, including mention of sugars, which may be key in synchronization.

Interestingly, coupling of clocks in different cells in plants as *Arabidopsis* through plasmodesmata increases synchrony of circadian rhythms and the identity of cell-to-cell signals coupling clocks has not been identified either (Greenwood *et al.* PLOS Biol. 17:e3000407 (2019)).

2. The physiological relevance of the mechanism described in this study is unclear to me. How do the authors envision such clock coupling provides robustness and a biological advantage under stress conditions? Could such robustness be assessed under various sugar availabilities or under starvation for example?

Synchronization of clocks along a filament through coupling endows the filament with the property to function as an organismic unit, allowing it to respond as a unit to external cues that promote responses at the gene expression level. The opposite situation, i.e., one in which every cell displays a different phase could lead to discoordinated gene expression responses, and consequently, to different fitness in each cell. In the more extreme situation, big differences in transcriptional responses could lead to parts of the filament being viable and others that are not. A sentence was revised in the Discussion about this issue (line 322). In the particular case of responses to nitrogen scarcity, a coordinated transcriptional response along the filament could be required for the establishment of a spatial pattern of nitrogen-fixing cells (heterocysts) distribution along the filament. Note that even heterocyst-enriched fractions display circadian oscillations as demonstrated in bulk cultures by the work of Kushige et al. (2013). We added a sentence to the Discussion stating this fact (lines 382-384).

Reviewer #2:This is a nice paper on measurements on circadian oscillations in *Anabaena* filaments. There has been a lot of investigations on circadian rhythms in various systems. I am not an experimentalist and I do not work on circadian systems so it is hard for me to judge whether this paper presents fundamental new measurements on the circadian clock and its relations to the cell cycle. However, the paper appears very comprehensive and presents to the best of my knowledge new nice results.Let me concentrate on the model presented in Figure 5A. This is defined by different interactions between various phosphorylated states of KaiC gene. I should like the authors to explain in more details the different links in this diagram. It probably makes sense that there are transitions between the phosphorylated states but please explain in more details. It is well know that oscillations are generated from negative feed-back loops.

We thank the reviewer for pointing this out. While the basic scheme was already introduced starting in line 206, we have defined the different symbols in the scheme in the figure legend. In addition, we have added to this panel the precise definition of the rates between the different phosphoforms of KaiC. Regarding the remark on feedback loops, please see our answer to the next comment.

In the diagram there are two negative links, from S state to Kai A and back again. This by itself defines a positive feed-back loop which will give rise to a switch not to oscillations. Therefore the only negative loop I can identify is the one S -> KaiA -> D(ST) -> S. Is that the loop that is responsible for all the oscillations? I might doubt it but please explain.

The crucial negative feedback loop in the core oscillator is mediated by S-KaiC, through inactivation of KaiA via KaiB function, as explained previously by Rust et al. 2007, who analyzed the deterministic model embodied by this diagram. We have introduced a short sentence with this statement in line 209 to make this clear. A detailed analysis of the molecular steps in this feedback loop has recently been reported (Hong et al. Mol. Sys. Biol. 16:e9355 (2020)).

We note in passing that the minimal core circuit represented in the diagram results in measurable oscillations, as demonstrated when only the KaiABC proteins are reconstituted *in vitro* together with ATP (Nakajima et al. Science 308:414 (2005)).

The authors have 'exported' all equations to the supporting information but I would have liked whether it is possible to include the equations for this basic loop as a diagram in the figure?

Length limitations prevent us from displaying the full deterministic equations, which have been presented previously (in Rust et al., Science 318:809 (2007)). However, following their work, we have added to Figure 5A the interconversion rates between KaiC phosphoforms X and Y, and their analytical dependence on active KaiA monomers, which itself depends on the concentration of the S-KaiC phosphoform. This dependence is different from that of Rust *et al.*

From the underlying deterministic model of this diagram the authors obtain the phase diagram in Figure 5C. I presume the line to the colored region is defined through a Hopf bifurcation, am I right? As the authors mention, there is 'only' a circadian time scale at the lower right boundary. But is that not surprising?

The reviewer is right. The boundary of the colored region corresponds to the onset of a Hopf bifurcation from a (deterministic) non-oscillating state to limit cycle oscillations (clearly reviewed in Markson, J.S., & O<milestone-start />’<milestone-end />Shea, E.K. *FEBS Letters*, *583*(24), 3938-3947 (2009)). We have made this clear in the revised legend of Figure 5. In the colored region, the period of oscillations varies with variations in parameter values. We do not deem this variation surprising; for different fixed points the oscillation period may be different.

Next, the authors presents a model for an array of coupled circadian clocks which is supposed to model *Anabaena* through cell-cell communications via septal proteins. The full structure of the model is nicely presented in the supporting information. Well, it is not a simple model that is developed and I cannot claim I understand all the involved steps. Some of the results are presented in Figure 6. Figure 6B shows results from Gillespie on how the cells are correlated. Bit I am missing a little more explanation. What is the noise level (volume/number of molecules in Gillespie?), max/min of what. And in the text there is an error where Figure 6B says it is the power spectrum. Altogether, I find it a nice paper.

We thank the reviewer for pointing out the lack of information relative to Figure 6B. Figure 6B reports the concentration of T-KaiC in a filament of 10 cells obtained from a single realization of the Gillespie algorithm run for 16 million steps. The strength of noise is controlled by the parameter N, the total number of KaiC phosphoforms, that we have set to 5000. We have included all those relevant details in the caption of Figure 6B. The typo in the text has been corrected: from Figure 6B to Figure 6C.

Reviewer #3:The system is very interesting and offers a rare natural example of a system of coupled oscillators which have been the subject of much theoretical investigation.I think the results certainly merit consideration in eLife. However, I think some of the theoretical claims made are not sufficiently supported in the paper in its current version.1. A central argument that the system does not display deterministic oscillations seems to rely heavily on Figure 5C. However, it is a little difficult to see how much that conclusion is robust to the very specific model and parameters used. For example, it would seem that a factor of 2 in [Kai C] and a factor of 10% in [KaiA] would completely change that conclusion. Surely, this is possible given that there is evidence that *Anabaena* differs from *Synechococcus* as the authors themselves claim. Perhaps there is no parameter combination (including parameters other than the two the authors chose to vary) which would show deterministic oscillations without making the Figure 5B fit significantly worse but this is a little unclear from the current text.

We thank the reviewer for raising this important point. In the former version of the paper, when carrying out the fit, all parameters (except for γ and the concentration [KaiA]) were frozen to the nominal values as determined in Rust *et al.* for the reference case of *Synechococcus*. Following the reviewer’s remark, we have implemented a bootstrap procedure to account for a degree of variability of the aforementioned kinetic parameters. More precisely, each kinetic parameter (k) is randomly selected from a Gaussian distribution centered at the nominal value k¯ estimated by Rust *et al.* (2007). The standard deviation of the distribution is assigned as: σ=k¯10. For every complete set of (randomly selected) kinetic constants, we proceed with a two-parameter fit (γ and [KaiA] let free to change) to interpolate the experimental power spectrum. Each pair of fitted values is stored to compute eventually averaged estimates, together with the error, as quantified by the associated standard deviation. By averaging over 200 independent realizations of the implemented procedure yields γ=7.2±1.4 (mean ± std) and [KaiA] =1.3±0.24, with a degree of the variability which is hence quantified in approximately 20%. It is worth emphasizing that each fit rests on a stochastic description of the dynamics. The bifurcation line that sets the separation between the deterministic limit cycle and the stationary stable fixed point is modulated, depending on the set of assigned kinetic constants. Each pair of parameters γ and [KaiA] from the fit falls by definition in the domain were deterministic oscillations do not occur and the stochastic finite size corrections drive the emergence of the resonant quasi cycles.

Our results suggests that circadian clocks could have a stochastic origin. The broad power spectrum profile as recorded in the experiments is a clear indication that noise plays a role, and available data are compatible with a stochastic cause for the onset of the observed oscillation. We have added a detailed discussion of this point from line 258 and a subsection at the end of Materials and methods explaining these procedures (line 517).

2. Following on that point, the authors do not seem to sufficiently discuss the applicability of Rust et al. model and parameters to *Anabaena*. I think the paper would be considerably strengthened by a clear discussion on how robust the conclusion is to different parameter variations and the applicability of this choice, as well as the possibility of deterministic vs noise driven oscillations.

To answer this point we have refined the analysis, by modulating the parameters as compared to their reference values estimated by Rust et al. The adopted procedure has been discussed above. For all sets of kinetic parameters, the experimental power spectrum is nicely interpolated by the corresponding (fitted) theoretical profile, which assumes that the observed oscillations have a stochastic origin. Based on this, we argue that circadian oscillations could result from a resonant amplification of the endogenous component of noise, as stemming from demographic fluctuations. As a matter of fact, we do not claim that circadian clocks are definitely stochastic oscillators. We only cast on solid grounds the observation that they could be adequately explained by resorting to a stochastic scenario, that successfully captures experimental data. In this regard, it is worth stressing that a fully deterministic mechanism for cycle generation would yield a δ-like profile in the frequency domain, in stark contrast with the broad spectrum that we have experimentally recorded.

3. There are three different Kernels used for the cell-to-cell communication, about which little is known experimentally. Of these, the exponential seems most plausible in the absence of other knowledge. The caption of Figure 6D says the complex coherence function is fit using an exponential kernel but the sharp cutoff in the fit seems to suggest a constant Kernel with a sharp cutoff. The SI suggests a power law Kernel fits well but it is unclear what the justification of such a Kernel would be. As an aside, is it possible to simplify the equations in a mean-field kind of way?

The model that we propose is stochastic in nature and the mean-field approximation yields the set of coupled ordinary differential equations that govern the dynamical evolution of the interacting species in the deterministic limit. In Figure 6D we analyze the fluctuation beyond the mean field limit, so as to bring into evidence the role played by fluctuations in cell-cell communication. Unfortunately, we could not find any viable strategy to simplify further the equations that yield the complex coherence function, under different couplings. However, we agree with the referee that further insight would be useful to discriminate eventually between the different kernels proposed.

4. It is a little unclear in Table 1 why R is so high even for different filaments particularly since it was originally introduced in Garcia-Ojalvo as an order parameter that sharply distinguishes between synchronization and lack of.

As detailed in the Materials and methods section, filaments are initially grown in flasks under specific conditions of light intensity and spectrum, prior to being put in a device for microscope, long-term observation. The process of putting filaments in devices and in the microscope setup involves manifold changes in illumination properties, temperature and adjustment to the constrained space of the device, where filaments are fed exclusively by nutrients in the gel pad. All these changes induce synchronization between filaments, yielding large values of R.

I think the paper is not suitable for eLife in its current form but a clearer and stronger discussion of model choice and fitting could remedy that.

It is our hope that the changes we have introduced in the revised manuscript have remedied the problems highlighted by the reviewer.